# Optimal and Fair Encouragement Policy Evaluation and Learning

**Angela Zhou**
Department of Data Sciences and Operations
University of Southern California
`zhoua@usc.edu`

## Abstract

In consequential domains, it is often impossible to compel individuals to take treatment, so that optimal treatment assignments are merely suggestions when humans make the final treatment decisions. On the other hand, there can be different heterogeneity in both the actual response to treatment and final treatment decisions given recommendations. For example, in social services, a persistent puzzle is the gap in take-up of beneficial services among those who may benefit from them the most. When decision-makers have equity- for fairness-minded preferences over both access and average outcomes, the optimal decision rule changes due to these differing heterogeneity patterns. We study identification and improved/robust estimation under potential violations of positivity. We consider fairness constraints such as demographic parity in treatment take-up, and other constraints, via constrained optimization. We develop a two-stage, online learning-based algorithm for solving over parametrized policy classes under general constraints to obtain variance-sensitive regret bounds. Our framework can be extended to handle algorithmic recommendations under an often-reasonable covariate-conditional exclusion restriction, using our robustness checks for lack of positivity in the recommendation.

## 1 Introduction

The intersection of causal inference and machine learning for heterogeneous treatment effect estimation can improve public health, increase revenue, and improve outcomes by personalizing treatment decisions, such as medications, e-commerce platform interactions, and social interventions, to those who benefit from it the most [4, 30, 35, 49]. But, in many important settings, we do not have direct control over treatment, and can only optimize over *encouragements*, or *recommendations* for treatment. For example, in e-commerce, companies can rarely *compel* users to sign up for certain services, rather *nudge* or *encourage* users to sign up via promotions and offers. When we are interested in optimizing the effects of signing up – or other voluntary actions beyond a platform's control – on important final outcomes such as revenue, we therefore need to consider *fairness-constrained optimal encouragement designs*. Often human expert oversight is required in the loop in important settings where ensuring *fairness in machine learning* [6] is also of interest: doctors prescribe treatment from recommendations [31], managers and workers combine their expertise to act based on decision support [8], and in the social sector, caseworkers assign to beneficial programs based on recommendations from risk scores that support triage [14, 19, 48].

The human in the loop requires new methodology for optimal encouragement designs because

> when the human in the loop makes the final prescription, algorithmic recommendations do not have direct causal effects on outcomes; they change the probability of treatment assignment.

37th Conference on Neural Information Processing Systems (NeurIPS 2023).

On the other hand, this is analogous to the well-understood notion of *non-compliance/non-adherence* in randomized controlled trials in the real world [22, 21]. For example, patients who are prescribed treatment may not actually take medication. A common strategy is to conduct an *intention-to-treat* analysis: under assumptions of no unobserved confounders affecting treatment take-up and outcome, we may simply view encouragement as treatment. But, in the case of prediction-informed decisions in social settings, if we are concerned about *access to the intervention* in addition to *utility of the policy over the population*, finer-grained analysis is warranted. If an outcome-optimal policy results in wide disparities in access, for example in marginalized populations not taking up incentives for healthy food due to lack of access in food deserts, or administrative burden that screens out individuals applying for social services that could benefit the most, this could be a serious concern for decision-makers. We ultimately may seek optimal decision rules that improve disparities in treatment access. In contrast, previous work in algorithmic accountability primarily focuses on auditing *recommendations*, but not both the access and efficacy achieved under the final decision rule. Therefore, previous methods can fall short in mitigating potential disparities.

Our contributions are as follows: we characterize optimal and resource fairness-constrained optimal decision rules, develop statistically improved estimators and robustness checks for the setting of algorithmic recommendations with sufficiently randomized decisions. We also develop methodology for optimizing over a constrained policy class with less conservative out-of-sample fairness constraint satisfaction by a two-stage procedure, and we provide sample complexity bounds. We assess improved recommendation rules in a stylized case study of optimizing recommendation of supervised release in the PSA-DMF pretrial risk-assessment tool while reducing surveillance disparities.

## 2 Related Work

In the main text, we briefly highlight the most relevant methodological and substantive work and defer additional discussion to the appendix.

**Optimal encouragement designs/policy learning with constraints.** There is extensive literature on off-policy evaluation and learning, empirical welfare maximization, and optimal treatment regimes [5, 49, 35, 30]. [39] studies an optimal individualized encouragement design, though their focus is on optimal individualized treatment regimes with instrumental variables. [27] study fairness in pricing, and some of the desiderata in that setting on revenue (here, marginal welfare) and demand (take-up) are again relevant here, but in a more general setting beyond pricing. The most closely related work in terms of problem setup is the formulation of "optimal encouragement designs" in [39]. However, they focus on knapsack resource constraints, which have a different solution structure than fairness constraints. Their outcome models in regression adjustment are conditional on the recommended/not recommended partitions which would not allow our fairness constraints that introduce treatment- and group-dependent costs. [44] has studied uniform feasibility in constrained resource allocation, but without encouragement or fairness. [9] studies robust extrapolation in policy learning from algorithmic recommendation, but not fairness. Our later case study is on supervised release, where there is a lot of randomness in final treatment decisions, rather than pretrial detention.

**Fair off-policy learning** We highlight some most closely related works in off-policy learning (omitting works in the sequential setting). [37] studies high-probability fairness constraint satisfaction. [29] studies doubly-robust causal fair classification, while others have imposed deterministic resource constraints on the optimal policy formulation [13]. [26] studies (robust) bounds for treatment responders in binary outcome settings; this desiderata is coupled to classification notions of direct treatment. Again, our focus is on modeling the fairness implications of non-adherence. Indeed, in order to provide general algorithms and methods, we do build on prior fair classification literature. A different line of work studies "counterfactual" risk assessments which models a different concern.

**Other causal methodology for intention-to-treat** We focus on deriving estimators for intention-to-treat analyses in view of fairness constraints (which result in group-specific welfare weights). Our interest is in imposing separate desiderata on treatment realizations under non-compliance; but we don't conduct instrumental variable inference and we assume unconfoundedness holds. We argue ITT is policy-relevant whereas complier-strata specific analysis is less policy-relevant since the compliers are unknown. Since our primary interest is in characterizing fair optimal decision rules, we don't model this as a mediation analysis problem (isolating the impact of recommendation even under the same ultimate treatment in a nested counterfactual), which may be more relevant for descriptive

characterization. [27] studies multi-objective desiderata for pricing and notes intention-to-treat structure in pricing, but not fairness considerations in more general problems. A related literature studies principal stratification [24], which has similar policy-relevance disadvantages regarding interpretability as complier analysis does.

## 3 Problem Setup

We briefly describe the problem setup. We work in the Neyman-Rubin potential outcomes framework for causal inference [40]. We define the following:

- recommendation flag $R \in \{0, 1\}$, where $R = 1$ means encouraged/recommended. (We will use the terms encouragement/recommendation interchangeably).
- treatment $T(R) \in \mathcal{T}$, where $T(r) = 1$ indicates the treatment decision was $1$ when the recommendation reported $r$.
- outcome $Y(t(r))$ is the potential outcome under encouragement $r$ and treatment $t$.

Regarding fairness, we will be concerned about disparities in utility and treatment benefits (resources or burdens) across different groups, denoted $A \in \{a, b\}$. (For notational brevity, we may generically discuss identification/estimation without additionally conditioning on the protected attribute). For example, recommendations arise from binary high-risk/low-risk labels of classifiers. In practice, in consequential domains, classifier decisions are rarely automated, rather used to inform humans in the loop. The human expert in the loop decides whether or not to assign treatment. For binary outcomes, we will interpret $Y(t(r)) = 1$ as the positive outcome, and when treatments are also binary, we may further develop analogues of fair classification criteria. We let $c(r, t, y) \colon \{0, 1\}^3 \mapsto \mathbb{R}$ denote the cost function for $r \in \{0, 1\}, t \in \mathcal{T}, y \in \{0, 1\}$, which may sometimes be abbreviated $c_{rt}(y)$. We discuss identification and estimation based on the following recommendation, treatment propensity, and outcome models:

$$e_r(X, A) \coloneqq P(R = r \mid X, A), \ \ p_{t|r}(X, A) \coloneqq P(T = t \mid R = r, X, A),$$
$$\mu_{r,t}(X, A) \coloneqq \mathbb{E}[c_{rt}(Y) \mid R = r, T = t, X, A] = \mathbb{E}[c_{rt}(Y) \mid T = t, X, A] \coloneqq \mu_t(X, A) \ \text{(asn.2)}$$

We are generally instead interested in *personalized recommendation rules* $\pi(r \mid X) = \pi_r(X)$ which describes the probability of assigning the recommendation $r$ to covariates $X$. The average encouragement effect is the difference in average outcomes if we refer everyone vs. no one, while the encouragement policy value $V(\pi)$ is the population expectation induced by the potential outcomes and treatment assignments realized under a recommendation policy $\pi$.

$$AEE = \mathbb{E}[Y(T(1)) - Y(T(0))], \qquad V(\pi) = \mathbb{E}[c(\pi, T(\pi), Y(\pi))].$$

Because algorithmic decision makers may be differentially responsive to recommendation, and treatment effects may be heterogeneous, the optimal recommendation rule may differ from the (infeasible) optimal treatment rule when taking constraints into account or for simpler policy classes.

**Assumption 1** (Consistency and SUTVA ). $Y_i = Y_i(T_i(R_i))$.

**Assumption 2** (Conditional exclusion restriction). $Y(T(R)) \perp\!\!\!\perp R \mid T, X, A$.

**Assumption 3** (Unconfoundedness). $Y(T(r)) \perp\!\!\!\perp T(r) \mid X, A$.

**Assumption 4** (Stable responsivities under new recommendations). $P(T = t \mid R = r, X)$ remains fixed from the observational to the future dataset.

**Assumption 5** (Decomposable costs). $c(r, t, y) = c_r(r) + c_t(t) + c_y(y)$

**Assumption 6** (Overlap). $\nu_r \leq e_r(X, A) \leq 1 - \nu_r; \ \ \nu_t \leq p_{t|r}(X, A) \leq 1 - \nu_t; \nu_r, \nu_t \leq 0$

Our key assumption beyond standard causal inference assumptions is the conditional exclusion restriction assumption 2, i.e. that conditional on observable information $X$, the recommendation has no causal effect on the outcome beyond its effect on increasing treatment probability. This assumes that all of the covariate information that is informative of downstream outcomes is measured. Although this may not exactly hold in all applications, stating this assumption is also a starting point for sensitivity analysis under violations of it [25]. Assuming assumption 6 is like assuming

we consider a randomized controlled trial with nonadherence. But later we give arguments using robustness to go beyond this, leveraging our finer-grained characterization.

Assumption 4 is a structural assumption that limits our method to most appropriately re-optimize over small changes to existing algorithmic recommendations. This is also required for the validity of intention-to-treat analyses. For example, $p_{0|1}(x)$ (disagreement with algorithmic recommendation) could be a baseline algorithmic aversion. Not all settings are appropriate for this assumption. We don't assume micro-foundations on how or why human decision-makers were deviating from algorithmic recommendations, but take these patterns as given. One possibility for relaxing this assumption is via conducting sensitivity analysis, i.e. optimizing over unknown responsivity probabilities near known ones.

Later on, we will be particularly interested in constrained formulations on the intention-to-treat effect that impose separate desiderata on outcomes under treatment, as well as treatment.

## 4  Method

We consider two settings: in the first, $R$ is (as-if) randomized and satisfies overlap. Then $R$ can be interpreted as intention to treat or prescription, whereas $T$ is the actual realization thereof. We study identification of optimal encouragement designs with potential constraints on treatment or outcome utility patterns by group membership. We characterize optimal unconstrained/constrained decisions under resource parity. In the second, $R$ is an algorithmic recommendation that does not satisfy overlap in recommendation (but there is sufficient randomness in human decisions to satisfy overlap in treatment): we derive robustness checks in this setting by being robust. First we discuss causal identification in optimal encouragement designs.

**Proposition 1** (Regression adjustment identification)**.**

$$\mathbb{E}[c(\pi, T(\pi), Y(\pi))] = \sum_{t \in \mathcal{T}, r \in \{0,1\}} \mathbb{E}[\pi_r(X)\mu_t(X)p_{t|r}(X)]$$

*Proof of Proposition 1.*

$$
\begin{aligned}
\mathbb{E}[c(\pi, T(\pi), Y(\pi))] &= \sum_{t \in \mathcal{T}, r \in \{0,1\}} \mathbb{E}[\pi_r(X)\mathbb{E}[\mathbb{I}\left[T(r) = t\right] c_{rt}(Y(r,t)) \mid R = r, X]] \\
&= \sum_{t \in \mathcal{T}, r \in \{0,1\}} \mathbb{E}[\pi_r(X)P(T = t \mid R = r, X)\mathbb{E}[c_{rt}(Y(r,t)) \mid R = r, X]] \\
&= \sum_{t \in \mathcal{T}, r \in \{0,1\}} \mathbb{E}[\pi_r(X)P(T = t \mid R = r, X)\mathbb{E}[c_{rt}(Y) \mid T = t, X]]
\end{aligned}
$$

where the last line follows by the conditional exclusion restriction (Assumption 2) and consistency (Assumption 1). $\square$

**Resource-parity constrained optimal decision rules**  We consider a resource/burden parity fairness constraint:

$$V_\epsilon^* = \max_\pi \left\{ \mathbb{E}[c(\pi, T(\pi), Y(\pi))] \colon \mathbb{E}[T(\pi) \mid A = a] - \mathbb{E}[T(\pi) \mid A = b] \leq \epsilon \right\} \tag{1}$$

Enforcing absolute values, etc. follows in the standard way. Not all values of $\epsilon$ may be feasible; in the appendix we give an auxiliary program to compute feasible ranges of $\epsilon$. We first characterize a threshold solution when the policy class is unconstrained.

**Proposition 2** (Threshold solutions)**.**  Define

$$L(\lambda, X, A) = (p_{1|1}(X, A) - p_{1|0}(X, A)) \left\{ \tau(X, A) + \frac{\lambda}{p(A)}(\mathbb{I}\left[A = a\right] - \mathbb{I}\left[A = b\right]) \right\} + \lambda(p_{1|0}(X, a) - p_{1|0}(X, b))$$

$$\lambda^* \in \arg\min_\lambda \mathbb{E}[L(\lambda, X, A)_+], \quad \pi^*(x, u) = \mathbb{I}\{L(\lambda^*, X, u) > 0\}$$

If instead $d(x)$ is a function of covariates $x$ only,

$$\lambda^* \in \arg\min_\lambda \mathbb{E}[\mathbb{E}[L(\lambda, X, A) \mid X]_+], \quad \pi^*(x) = \mathbb{I}\{\mathbb{E}[L(\lambda^*, X, A) \mid X] > 0\}$$

Establishing this threshold structure (follows by duality of infinite-dimensional linear programming) allows us to provide a generalization bound argument.

## 4.1 Generalization

**Proposition 3** (Policy value generalization)**.** Assume the nuisance models $\eta = [p_{1|0}, p_{1|1}, \mu_1, \mu_0]^\top, \eta \in H$ are consistent and well-specified with finite VC-dimension $V_\eta$ over the product function class $H$. Let $\Pi = \{\mathbb{I}\{\mathbb{E}[L(\lambda, X, A; \eta) \mid X] > 0 : \lambda \in \mathbb{R}; \eta \in \mathcal{F}\}$.

$$\sup_{\pi \in \Pi, \lambda \in \mathbb{R}} |(\mathbb{E}_n[\pi L(\lambda, X, A)] - \mathbb{E}[\pi L(\lambda, X, A)])| = O_p(n^{-\frac{1}{2}})$$

This bound is stated for known nuisance functions: verifying stability under estimated nuisance functions further requires rate conditions.

**Doubly-robust estimation**   We may improve statistical properties of estimation by developing *doubly robust* estimators which can achieve faster statistical convergence when both the probability of recommendation assignment (when it is random), and the probability of outcome are consistently estimated; or otherwise protect against misspecification of either model. We first consider the ideal setting when algorithmic recommendations are randomized so that $e_r(X) = P(R = r \mid X)$.

**Proposition 4** (Variance-reduced estimation)**.**

$$V(\pi) = \sum_{t \in \mathcal{T}, r \in \{0,1\}} \mathbb{E}\left[\pi_r(X)\left\{\frac{\mathbb{I}[R=r]}{e_r(X)}(\mathbb{I}[T=t]c_{r1}(Y) - \mu_1(X)p_{t|r}(X)) + \mu_1(X)p_{t|r}(X)\right\}\right]$$

$$\mathbb{E}[T(\pi)] = \sum_{r \in \{0,1\}} \mathbb{E}\left[\pi_r(X)\left\{\frac{\mathbb{I}[R=r]}{e_r(X)}(T(r) - p_{1|r}(x)) + p_{1|r}(x)\right\}\right]$$

Although similar characterization appears in [39] for the doubly-robust policy value alone, note that doubly-robust versions of the constraints we study would result in differences in the Lagrangian so we retain the full expression rather than simplifying. For example, for regression adjustment, Proposition 9 provides interpretability on how constraints affect the optimal decision rule. In the appendix we provide additional results describing extensions of Proposition 8 with improved estimation.

## 4.2 Robust estimation with treatment overlap but not recommendation overlap

When recommendations are e.g. the high-risk/low-risk labels from binary classifiers, we may not satisfy the overlap assumption, since algorithmic recommendations are deterministic functions of covariates. However, note that identification in Proposition 1 requires only SUTVA and consistency, and the exclusion restriction assumption. Additional assumptions may be required to extrapolate $p_{t|r}(X)$ beyond regions of common support. On the other hand, supposing that positivity held with respect to $T$ given covariates $X$, given unconfoundedness, our finer-grained approach can be beneficial because we only require robust extrapolation of $p_{t|r}(X)$, response to recommendations, rather than the outcome models $\mu_t(X)$.

We first describe what can be done if we allow ourselves parametric extrapolation on $p_{1|1}(X)$, treatment responsivity. In the case study later on, the support of $X \mid R = 1$ is a superset of the support of $X \mid R = 0$ in the observational data. Given this, we derive the following alternative identification based on marginal control variates (where $p_t = P(T = t \mid X)$ marginalizes over the distribution of $R$ in the observational data):

**Proposition 5** (Control variate for alternative identification )**.** Assume that $Y(T(r)) \perp T(r) \mid R = r, X$.

$$V(\pi) = \sum_{t \in \mathcal{T}, r \in \{0,1\}} \mathbb{E}\left[\left\{c_{rt}(Y(t))\frac{\mathbb{I}[T=t]}{p_t(X)} + \left(1 - \frac{\mathbb{I}[T=t]}{p_t(X)}\right)\mu_t(X)\right\}p_{t|r}(X)\right]$$

**Robust extrapolation of $p_{t|r}(X)$**   Let $\mathcal{X}^{\text{no}} = \{x : P(R = 1 \mid X) = 0\}$ denote the no overlap region; on this region there are no joint observations of $(t, r, x)$. We consider uncertainty sets for ambiguous treatment recommendation probabilities. For example, one plausible structural assumption is *monotonicity*, that is, making an algorithmic recommendation can only increase the probability of being treated.

We define the following uncertainty set:

$$\mathcal{U}_{q_{t|r}} := \left\{ q_{1|r}(x') \colon q_{1|r}(x) \geq p_{1|r}(x),\ \forall x \in \mathcal{X}^{\mathrm{no}}, r\ \textstyle\sum_{t\in\mathcal{T}} q_{t|r}(x) = 1, \forall x, r \right\}$$

We could assume uniform bounds on unknown probabilities, or more refined bounds, such as Lipschitz-smoothness with respect to some distance metric $d$, or boundedness.

$$\mathcal{U}_{\mathrm{lip}} := \left\{ q_{1|r}(x') \colon d(q_{1|r}(x'), p_{1|r}(x)) \leq L d(x', x),\ (x', x) \in (\mathcal{X}^{\mathrm{no}} \times \mathcal{X}^{\mathrm{no}}) \right\}, \mathcal{U}_{\mathrm{bnd}} := \left\{ q_{1|r}(x') \colon \underline{b}(x) \leq q_{1|r}(x') \leq \overline{b}(x) \right\}$$

Define $V_{ov}(\pi) := \sum_{t\in\mathcal{T}, r\in\{0,1\}} \mathbb{E}[\pi(r \mid X) p_{t|r}(X) \mu_t(X) \mathbb{I}_{ov}]$. Let $\mathcal{U}$ denote the uncertainty set including any custom constraints, e.g. $\mathcal{U} = \mathcal{U}_{q_{1|r}} \cap \mathcal{U}_{\mathrm{lip}}$. For brevity we use $\mathbb{I}_{no}$ to denote $\mathbb{I}[X \in \mathcal{X}^{\mathrm{no}}]$, $\mathbb{I}_{ov}$ to denote $\mathbb{I}[X \in \mathcal{X}^{\mathrm{ov}}]$. Then we may obtain robust bounds by optimizing over regions of no overlap:

$$\overline{V}(\pi) := V_{ov}(\pi) + \overline{V}_{no}(\pi),\ \ \overline{V}_{no}(\pi) := \max_{q_{tr}(X)\in\mathcal{U}} \left\{ \textstyle\sum_{t\in\mathcal{T}, r\in\{0,1\}} \mathbb{E}[\pi(r \mid X) \mu_t(X) q_{tr}(X) \mathbb{I}_{no}]] \right\}$$

In the specialized, but practically relevant case of binary outcomes/treatments/recommendations, we obtain the following simplifications for bounds on the policy value, and the minimax robust policy that optimizes the worst-case overlap extrapolation function. In the special case of constant uniform bounds, it is equivalent (in the case of binary outcomes) to consider marginalizations:

**Lemma 1** (Binary outcomes, constant bound)**.** *Let* $\mathcal{U}_{cbnd} := \left\{ q_{t|r}(x') \colon \underline{B} \leq q_{1|r}(x') \leq \overline{B} \right\}$ *and* $\mathcal{U} = \mathcal{U}_{q_{t|r}} \cap \mathcal{U}_{cbnd}$. *Define* $\beta_{t|r}(a) := \mathbb{E}[q_{t|r}(X, A) \mid T = t, A = a]$. *If* $T \in \{0,1\}$,

$$\overline{V}_{no}(\pi) = \sum_{t\in\mathcal{T}, r\in\{0,1\}} \mathbb{E}[c_{rt}^* \beta_{t|r} \mathbb{E}[Y\pi(r \mid X) \mid T = t]\mathbb{I}_{no}]],$$

*where* $c_{rt}^* = \begin{cases} \overline{B}\,\mathbb{I}[t=1] + \underline{B}\,\mathbb{I}[t=0] & \text{if } \mathbb{E}[Y\pi(r \mid X) \mid T = t] \geq 0 \\ \overline{B}\,\mathbb{I}[t=0] + \underline{B}\,\mathbb{I}[t=1] & \text{if } \mathbb{E}[Y\pi(r \mid X) \mid T = t] < 0 \end{cases}$

We state the next result for simple uncertainty sets, like intervals, to deduce insights about the robust policy. In the appendix we include a more general reformulation for polytopic uncertainty sets.

**Proposition 6** (Robust linear program )**.** Suppose $R, T \in \{0,1\}$, and $q_{r1}(\cdot, u) \in \mathcal{U}_{bnd}, \forall r, u$. Define

$$\tau(x, a) = \mu_1(x, a) - \mu_0(x, a),\ \ \Delta B_r(x, u) = (\overline{B}_r(x, u) - \underline{B}_r(x, u)), B_r^{\mathrm{mid}}(x, u) = \underline{B}_r(x, u) + \frac{1}{2}\Delta B_r(x, u),$$

$$\mathbb{E}[\Delta_{ov} T(\pi)] = \mathbb{E}[T(\pi)\mathbb{I}_{ov} \mid A = a] - \mathbb{E}[T(\pi)\mathbb{I}_{ov} \mid A = b],\ \ c_1(\pi) = \textstyle\sum_r \mathbb{E}[\tau_r \pi_r B^{\mathrm{mid}}]$$

Then the robust linear program is:

$$\max V_{ov}(\pi) + \mathbb{E}[\mu_0] + c_1(\pi) - \tfrac{1}{2}\textstyle\sum_r \mathbb{E}[|\tau|\,\pi(r \mid X)\Delta B_r(X, A)\mathbb{I}_{no}]$$
$$\text{s.t. } \textstyle\sum_r \{\mathbb{E}[\pi(r \mid X)\overline{B}_r(X, A)\mathbb{I}_{no} \mid A = a] - \mathbb{E}[\pi_r \underline{B}_r(X, A)\mathbb{I}_{no} \mid A = b]\} + \Delta_{ov}^T(\pi) \leq \epsilon$$

## 5 Additional fairness constraints and policy optimization

We previously discussed policy optimization, over unrestricted decision rules, given estimates. We now introduce general methodology to handle 1) optimization over a policy class of restricted functional form and 2) more general fairness constraints. We first introduce the fair-classification algorithm of [1], describe our extensions to obtain variance-sensitive regret bounds and less conservative policy optimization (inspired by a regularized ERM argument given in [12]), and then provide sample complexity analysis.

**Algorithm and setup**   In the following, to be consistent with standard form for linear programs, note that we consider outcomes $Y$ to be costs so that we can phrase the saddle-point as minimization-maximization. Consider $|\mathcal{K}|$ linear constraints and $J$ groups (values of protected attribute $A$), with coefficient matrix $M \in \mathbb{R}^{K \times J}$, the constraint moment function $h_j(\pi), j \in [J]$ (with $J$ the number of groups), $O = (X, A, R, T, Y)$ denoting our data observations, and $d$ the constraint constant vector:

$$h_j(\pi) = \mathbb{E}\left[g_j(O, \pi(X)) \mid \mathcal{E}_j\right] \quad \text{for } j \in J,\ \ Mh(\pi) \leq d$$

Importantly, $g_j$ depends on $\pi$ while the conditioning event $\mathcal{E}_j$ cannot depend on $\pi$. Many important fairness constraints can nonetheless be written in this framework, such as burden/resource parity,

**Algorithm 1** MW2REDFAIR($\mathcal{D}, g, \mathcal{E}, M, d$)

---

1: Input: $\mathcal{D} = \{(X_i, R_i, T_i, Y_i, A_i)\}_{i=1}^{n}$, $g, \mathcal{E}, M, \widehat{d}$, B, $\nu$, $\alpha$, $\theta_1 = 0 \in \mathbb{R}^{|\mathcal{K}|}$
2: **for** $t = 1, 2, \ldots$ **do**
3:     Set $\lambda_{t,k} = B \frac{\exp\{\theta_k\}}{1 + \sum_{k' \in \mathcal{X}} \exp\{\theta_{k'}\}}$ for all $k \in \mathcal{K}$, $\beta_t \leftarrow \text{BEST}_\beta(\lambda_t)$, $\widehat{Q}_t \leftarrow \frac{1}{t} \sum_{t'=1}^{t} \beta_{t'}$

       $\bar{L} \leftarrow L(\widehat{Q}_t, \text{BEST}_\lambda(\widehat{Q}_t))$, $\hat{\lambda}_t \leftarrow \frac{1}{t} \sum_{t'=1}^{t} \lambda_{t'}$, $\underline{L} \leftarrow (\text{BEST}_\beta(\hat{\lambda}_t), \hat{\lambda}_t)$,
5:     $\nu_t \leftarrow \max\{L(\widehat{Q}_t, \hat{\lambda}_t) - \underline{L}, \quad \bar{L} - L(\widehat{Q}_t, \hat{\lambda}_t)\}$, If $\nu_t \leq \nu$ then return $(\widehat{Q}_t, \hat{\lambda}_t)$
6:     $\theta_{t+1,i} = \theta_t + \log(1 - \eta(M\hat{\mu}(h_t) - \hat{c})_j), \forall i$
7: **end for**

---

parity in true positive rates, but not measures such as calibration whose conditioning event does depend on $\pi$. (See Appendix B.2 for examples omitted for brevity). We further consider a convexification of $\Pi$ via randomized policies $Q \in \Delta(\Pi)$, where $\Delta(\Pi)$ is the set of distributions over $\Pi$, i.e. a randomized classifier that samples a policy $\pi \sim Q$. Therefore we solve

$$\min_{Q \in \Delta(\Pi)} \{V(\pi): \quad Mh(\pi) \leq d\}$$

On the other hand, the optimization is solved using sampled moments, so that we ought to interpret the constraint values $\hat{d}_k = d_k + \epsilon_k$, for all $k$. The overall algorithmic scheme is similar: we seek an approximate saddle point so that the constrained solution is equivalent to the Lagrangian,

$$L(Q, \lambda) = \hat{V}(Q) + \lambda^\top (M\hat{h}(Q) - \hat{d}), \qquad \min_{Q \in \Delta(\Pi)} \{V(\pi): Mh(\pi) \leq d\} = \min_{Q \in \Delta(\Pi)} \max_{\lambda \in \mathbb{R}_+^K} L(Q, \lambda).$$

We simultaneously solve for an approximate saddle point and bound the domain of $\Lambda$ by $B$:

$$\min_{Q \in \Delta} \max_{\lambda \in \mathbb{R}_+^{|\mathcal{X}|}, \|\lambda\|_1 \leq B} L(Q, \lambda), \qquad \max_{\lambda \in \mathbb{R}_+^{|\mathcal{X}|}, \|\lambda\|_1 \leq B} \min_{Q \in \Delta} L(Q, \lambda)$$

We play a no-regret (second-order multiplicative weights [11, 43], a slight variant of Hedge/exponentiated gradient [18]) algorithm for the $\lambda-$player, while using best-response oracles for the $Q-$player. Full details are in Algorithm 1. Given $\lambda_t$, $\text{BEST}_\beta(\lambda_t)$ computes a best response over $Q$; since the worst-case distribution will place all its weight on one classifier, this step can be implemented by a reduction to cost-sensitive/weighted classification [10, 49], which we describe in further detail below. Computing the best response over $\text{BEST}_\lambda(\widehat{Q}_t)$ selects the most violated constraint. We include further details in Appendix B.2.

**Weighted classification reduction.** There is a well-known reduction of optimizing the zero-one loss for policy learning to weighted classification. Taking the Lagrangian will introduce datapoint-dependent additional weights. This reduction requires $\pi \in \{-1, +1\}, T \in \{-1, +1\}$ (for notational convenience alone). We consider parameterized policy classes so that $\pi(x) = \text{sign}(g_\theta(x))$ for some index function $g$ depending on a parameter $\beta \in \mathbb{R}^d$. Consider the centered regret $J(\pi) = \mathbb{E}[Y(\pi)] - \frac{1}{2}\mathbb{E}[\mathbb{E}[Y \mid R = 1, X] + \mathbb{E}[Y \mid R = 0, X]]$. Then $J(\beta) = J(\text{sgn}(g_\beta(\cdot))) = \mathbb{E}[\text{sgn}(g_\beta(X))\{\psi\}]$ where $\psi$ can be one of, where $\mu_r^R(X) = \mathbb{E}[Y \mid R = r, X]$,

$$\psi_{DM} = (p_{1|1}(X) - p_{1|0}(X))(\mu_1(X) - \mu_0(X)), \psi_{IPW} = \frac{RY}{e_R(X)}, \psi_{DR} = \psi_{DM} + \psi_{IPW} + \frac{R\mu^R(X)}{e_R(X)}$$

We can apply the standard reduction to cost-sensitive classification since $\psi_i \text{sgn}(g_\beta(X_i)) = |\psi_i|(1 - 2\mathbb{I}[\text{sgn}(g_\beta(X_i)) \neq \text{sgn}(\psi_i)])$. Then we can use surrogate losses for the zero-one loss. Although many functional forms for $\ell(\cdot)$ are Fisher-consistent, one such choice of $\ell$ is the logistic (cross-entropy) loss given below:

$$L(\beta) = \mathbb{E}[|\psi| \ell(g_\beta(X), \text{sgn}(\psi))], \qquad l(g, s) = 2\log(1 + \exp(g)) - (s + 1) \tag{2}$$

**Two-stage variance-constrained algorithm.** We seek to improve upon this procedure so that we may obtain regret bounds on policy value and fairness constraint violation that exhibit more favorable dependence on the maximal variance over small-variance *slices* near the optimal policy, rather than worst-case constants over all policies, [12, 5]. Further, note that the algorithm sets the constraint feasibility slacks via generalization bounds that previously depended on these worst-case constants.

---
**Algorithm 2** Two-stage localized fair classification via reductions
---
1: Randomly split the data into $\mathcal{D}_1, \mathcal{D}_2$
2: Obtain $\hat{\pi}_1^*$ and the index set of binding constraints $\hat{\mathcal{I}}_1$ by learning nuisances $\eta_1$ and running Algorithm 1 on $\mathcal{D}_1$ with MW2REDFAIR($\mathcal{D}_1, h, \mathcal{E}, M, d; \eta_1$)
3: $\hat{\sigma}_j^2 \leftarrow \hat{\mathrm{Var}}(g_j(O, \hat{\pi}_1^*) \mathbb{I}[\mathcal{E}_j]/p_j), \; \forall j; \; \hat{d} \leftarrow d + 2\hat{\sigma}^2 n^{-\alpha}$
4: Augment additional constraints with $\epsilon_n$-policy-value and constraint slices relative to $\hat{\pi}_1$: define an augmented system (where subscripting by $\hat{\mathcal{I}}_1$ indexes the corresponding matrix or vector):

$$\tilde{h}_{j'} = \mathbb{E}_{n_1}[\{g_{j'}^{\hat{\mathcal{I}}_1}(O; \hat{\pi}_1^*) - g_{j'}^{\hat{\mathcal{I}}_1}(O; \pi)\} \mid \mathcal{E}_j], \; \forall j' \in \hat{\mathcal{I}}_1, \; \tilde{h}^v = \mathbb{E}_{n_1}[v_{DR}(O; \pi, \eta_1) - v_{DR}(O; \pi, \eta_1)]$$
$$\tilde{M} = [M; M_{\hat{\mathcal{I}}_1}, \vec{1}], \tilde{h} = [h(\cdot; \pi), \tilde{h}, \tilde{h}^v]^\top, \tilde{d} = [\hat{d}, \epsilon_n, \epsilon_n]^\top, \tilde{\mathcal{E}} = [\mathcal{E}, \mathcal{E}_{\hat{\mathcal{I}}_1}, \emptyset]^\top$$

5: Obtain $\hat{\pi}_2^*$ by running Algorithm 1 on $\mathcal{D}_2$ with MW2REDFAIR($\mathcal{D}, \tilde{g}, \tilde{\mathcal{E}}, \tilde{M}, \tilde{d}, \eta_2$)

---

These challenges motivate the two-stage procedure, described formally in Algorithm 2 and verbally here. We adapt an out-of-sample regularization scheme developed in [12], which recovers variance-sensitive regret bounds via a small modification to ERM. We split the data, learn nuisance estimators $\eta_1$ for use in our policy value and constraint estimates, and run Algorithm 1 (MW2REDFAIR($\mathcal{D}_1, h, \mathcal{E}, M, d; \eta_1$)) to obtain an estimate of the optimal policy $\hat{\pi}_1$, and the constraint variances at $\hat{\pi}_1$. Next, we *augment* the constraint matrix with additional constraints that require the second-stage $\pi$ to achieve $\epsilon_n$ close policy value and constraint moment values relative to $\hat{\pi}_1$. Since errors concentrate fast, this can be viewed as variance regularization. And, we set the constraint slacks $\tilde{d}$ in the second stage using estimated variance constants from $\hat{\pi}_1$, which will be less conservative than using worst-case bounds. Define

$$v^{(\cdot)}(Q) = \mathbb{E}_{\pi \sim Q}[\pi \psi_{(\cdot)} \mid O], \text{ for } (\cdot) \in \{\emptyset, \mathrm{DR}\}, g_j(O; Q) = \mathbb{E}_{\pi \sim Q}[g_j(O; \pi) \mid O, \mathcal{E}_j],$$

so that $V^{(\cdot)}(Q) = \mathbb{E}[v^{(\cdot)}(Q)]$ and $h_j(Q) = \mathbb{E}[g_j(O; Q) \mid \mathcal{E}_j]$. Define the function classes

$$\mathcal{F}_\Pi = \{v_{DR}(\cdot, \pi; \eta) : \pi \in \Pi, \eta \in H\}, \mathcal{F}_j = \{g(\cdot, \pi; \eta) : \pi \in \Pi, \eta \in H\}$$

and the empirical entropy integral $\kappa(r, \mathcal{F}) = \inf_{\alpha \geq 0}\{4\alpha + 10 \int_\alpha^r \sqrt{\frac{\mathcal{H}_2(\epsilon, \mathcal{F}, n)}{n}} d\epsilon\}$ where $H_2(\epsilon, \mathcal{F}, n)$ is the $L_2$ empirical entropy, i.e. log of the $\|\cdot\|_2$ $\epsilon$-covering number. We make a mild assumption of a learnable function class (bounded entropy integral) [46].

**Assumption 7.** The function classes $\mathcal{F}_\Pi, \{\mathcal{F}_j\}_{j \in \mathcal{J}}$ satisfy that for any constant $r, \kappa(r, \mathcal{F}) \to 0$ as $n \to \infty$. The function classes $\{\mathcal{F}_j\}_j$ comprise of $L_j$-Lipschitz contractions of $\pi$.

We will assume that we are using doubly-robust/orthogonalized estimation as in proposition 4, hence state results depending on estimation error of nuisance vector $\eta$.

**Theorem 1** (Variance-Based Oracle Policy Regret). *Suppose that the mean-squared-error of the nuisance estimates is upper bounded w.p. $1 - \delta/2$ by $\chi_{n,\delta}^2$, over the randomness of the nuisance sample:* $\max_l\{\mathbb{E}[(\hat{\eta}_l - \eta_l)^2]\}_{l \in [L]} := \chi_n^2$. *Let $r = \sup_{Q \in \mathcal{Q}} \sqrt{\mathbb{E}[v_{DR}(z; \pi)^2]}$ and $\epsilon_n = \Theta(\kappa(r, \mathcal{F}_\Pi) + r\sqrt{\frac{\log(1/\delta)}{n}})$. Moreover, let*

$$\mathcal{Q}_*(\epsilon) = \{Q \in \Delta[\Pi] : V(Q_*^0) - V(Q) \leq \epsilon, \; h(Q_*^0) - h(Q) \leq d + \epsilon\}$$

*denote an $\epsilon$-regret slice of the policy space. Let $\tilde{\epsilon}_n = O(\epsilon_n + \chi_{n,\delta}^2)$ and*

$$V_2^0 = \sup \{\mathrm{Var}(v_{DR}^0(O; Q) - v_{DR}^0(O; Q')) : Q, Q' \in \mathcal{Q}_*(\tilde{\epsilon}_n)\}$$

*denote the variance of the difference between any two policies in an $\epsilon_n$-regret slice, evaluated at the true nuisance quantities. (Define $V_2^j$ analogously for the variance of constraint moments). Then, letting $\gamma(Q) := Mh(Q)$ denote the constraint values, the policy distribution $Q_2$ returned by the out-of-sample regularized ERM, satisfies w.p. $1 - \delta$ over the randomness of $S$ :*

$$V(\hat{Q}) - V(Q^*) = O(\kappa(\sqrt{V_2^{obj}}, \mathrm{conv}(\mathcal{F}_\Pi)) + n^{-\frac{1}{2}}\sqrt{V_2^{obj} \log(3/\delta)} + \chi_{n,\delta}^2)$$

$$(\gamma_j(\hat{Q}) - c_j) - (\gamma_j(Q^*) - c_j) = O(\kappa(\sqrt{V_2^j}, \mathrm{conv}(\mathcal{F}_j)) + n^{-\frac{1}{2}}\sqrt{V_2^j \log(3/\delta)} + \chi_{n,\delta}^2)$$

The benefits are that 1) the constants are improved from an absolute, structure-agnostic bound to depending on the variance of low-regret policies, which also reflects improved variance from using doubly-robust estimation as in proposition 4, and 2) less-conservative out-of-sample fairness constraint satisfaction.

## 6 Experiments

Due to space constraints, in the main text we only present a case study based on the PSA-DMF for supervised release [38]. In the appendix we include additional experiments and robustness checks, including a case study of fully-randomized recommendations and non-adherence.

**PSA-DMF case study.** Our case study is on a dataset of judicial decisions on *supervised* release based on risk-score-informed recommendations [38]. The PSA-DMF (Public Safety Assessment Decision Making Framework) uses a prediction of failure to appear for a future court data to inform pretrial decisions, including our focus on supervised release (i.e. electronic monitoring in addition to release) where judges make the final decision. Despite a large literature on algorithmic fairness of pretrial risk assessment, to the best of our knowledge, recommendations about supervised release are not informed by empirical evidence. There are current policy concerns about disparities in increasing use of supervised release given mixed evidence on outcomes [38]; e.g. Safety and Justice Challenge [41] concludes "targeted efforts to reduce racial disparities are necessary". First, we acknowledge data issues. We work with publicly available data that was discretized for privacy [38]. The final supervision decision does not include intensity, but different intensities are recommended in the data, which we collapse into a single level. The PSA-DMF is an algorithmic recommendation so here we are appealing to overlap in treatment recommendations, but using parametric extrapolation in responsivity. Finally, unconfoundedness is likely untrue, but sensitivity analysis could address this. So, this analysis should be considered exploratory, to illustrate the relevance of the methods. Future work will seek proprietary data for a higher-fidelity empirical study.

Next in Figure 1 we provide descriptive information illustrating heterogeneity (including by protected attribute) in adherence and effectiveness. We observe wide variation in judges assigning supervised release beyond the recommendation. We use logistic regression to estimate outcome models and treatment response models. The first figure shows estimates of the causal effect for different groups, by gender (similar heterogeneity for race). The outcome is failure to appear, so negative scores are beneficial. The second figure illustrates the difference in responsiveness: how much more likely decision-makers are to assign treatment when there is vs. isn't an algorithmic recommendation to do so. The last figure plots a logistic regression of the lift in responsiveness on the causal effect $\mu_1(x,a) - \mu_0(x,a)$. We observe disparities in how responsive decision-makers are conditional on the same treatment effect efficacy. This is importantly not a claim of animus because decision-makers didn't have access to causal effect estimates. Nonetheless, disparities persist.

In Figure 2 we highlight results from constrained policy optimization. The first two plots in each set illustrate the objective function value and $A = a$ average treatment cost, respectively; for $A$ being race (nonwhite/white) or gender (female/male), respectively. We use costs of $100$ for $Y = 1$ (failure to appear, $0$ for $Y = 0$, and $20$ when $T = 1$ (set arbitrarily). On the x-axis we plot the penalty $\lambda$ that we use to assess the solutions of Proposition 9. The vertical dashed line indicates the solution achieving $\epsilon = 0$, i.e. parity in treatment take-up. Near-optimal policies that reduce treatment disparity can be of interest due to advocacy concerns about how the expansion of supervised release could increase the surveillance of already surveillance-burdened marginalized populations. We see that indeed, for race, surveillance-parity constrained policies can substantially reduce disparities for nonwhites while not increasing surveillance on whites that much: the red line decreases significantly at low increase to the blue line (and low increases to the objective value). On the other hand, for gender, the opportunity for improvement in surveillance disparity is much smaller. See the appendix for further experiments and computational details.

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

# A   Additional discussion

## A.1   Additional related work

**Principal stratification and mediation analysis in causal inference**   [32] studies an optimal test-and-treat regime under a no-direct-effect assumption, that assigning a diagnostic test has no effect on outcomes except via propensity to treat, and studies semiparametric efficiency using Structural Nested-Mean Models. Though our exclusion restriction is also a no-direct-effect assumption, our optimal treatment regime is in the space of recommendations only as we do not have control over the final decision-maker, and we consider generally nonparametric models.

We briefly go into more detail about formal differences, due to our specific assumptions, that delineate the differences to mediation analysis. Namely, our conditional exclusion restriction implies that $Y_{1T_0} = Y_{T_0}$ and that $Y_{0T_1} = Y_{1T_1}$ (in mediation notation with $T_r = T(r)$ in our notation), so that so-called *net direct effects* are identically zero and the *net indirect effect* is the treatment effect (also called average encouragement effect here).

**Human-in-the-loop in consequential domains.**   There is a great deal of interest in designing algorithms for the "human in the loop" and studying expertise and discretion in human oversight in consequential domains [14]. On the algorithmic side, recent work focuses on frameworks for learning to defer or human-algorithm collaboration. Our focus is *prior* to the design of these procedures for improved human-algorithm collaboration: we primarily hold fixed current human responsiveness to algorithmic recommendations. Therefore, our method can be helpful for optimizing local nudges. Incorporating these algorithmic design ideas would be interesting directions for future work.

**Empirical literature on judicial discretion in the pretrial setting.**   Studying a slightly different substantive question, namely causal effects of pretrial decisions on later outcomes, a line of work uses individual judge decision-makers as a leniency instrumental variable for the treatment effect of (for example, EM) on pretrial outcomes [3, 2, 34]. And, judge IVs rely on quasi-random assignment of individual judges. We focus on the prescriptive question of optimal recommendation rules in view of patterns of judicial discretion, rather than the descriptive question of causal impacts of detention on downstream outcomes.

A number of works have emphasized the role of judicial discretion in pretrial risk assessments in particular [20, 15, 33]. In contrast to these works, we focus on studying decisions about electronic monitoring, which is an intermediate degree of decision lever to prevent FTA that nonetheless imposes costs. [23] study a randomized experiment of provision of the PSA and estimate (the sign of) principal causal effects, including potential group-conditional disparities. They are interested in a causal effect on the principal stratum of those marginal defendants who would not commit a new crime if recommended for detention. [9] study policy learning in the absence of positivity (since the PSA is a deterministic function of covariates) and consider a case study on determining optimal recommendation/detention decisions; however their observed outcomes are downstream of judicial decision-making. Relative to their approach, we handle lack of overlap via an exclusion restriction so that we only require ambiguity on *treatment responsivity models* rather than causal outcome models.

# B Additional discussion on method

## B.1 Additional discussion on constrained optimization

**Feasibility program** We can obtain upper/lower bounds on $\epsilon$ in order to obtain a feasible region for $\epsilon$ by solving the below optimization over maximal/minimal values of the constraint:

$$\bar{\epsilon}, \underline{\epsilon} \in \max_{\pi} / \min_{\pi} \mathbb{E}[T(\pi) \mid A = a] - \mathbb{E}[T(\pi) \mid A = b] \tag{3}$$

$$V_{\epsilon}^{*} = \max_{\pi} \left\{ \mathbb{E}[c(\pi, T(\pi), Y(\pi))] \colon \mathbb{E}[T(\pi) \mid A = a] - \mathbb{E}[T(\pi) \mid A = b] \leq \epsilon \right\} \tag{4}$$

## B.2 Additional discussion on Algorithm 2 (general algorithm)

### B.2.1 Additional fairness constraints and examples in this framework

In this section we discuss additional fairness constraints and how to formulate them in the generic framework. Much of this discussion is quite similar to [1] (including in notation) and is included in this appendix for completeness only. We only additionally provide novel identification results for another fairness measure on causal policies in Appendix B.2.2, concrete discussion of the reduction to weighted classification, and provide concrete descriptions of the causal fairness constraints in the more general framework.

We first discuss how to impose the treatment parity constraint. This is similar to the demographic parity example in [1], with different coefficients, but included for completeness. (Instead, recommendation parity in $\mathbb{E}[\pi \mid A = a]$ is indeed nearly identical to demographic parity.)

**Example 1** (Writing treatment parity in the general constrained classification framework.)**.** We write the constraint

$$\mathbb{E}[T(\pi) \mid A = a] - \mathbb{E}[T(\pi) \mid A = b] \tag{5}$$

in this framework as follows:

$$\mathbb{E}[T(\pi) \mid A = a] = \mathbb{E}[\pi_1(X)(p_{1|1}(X, A) - p_{1|0}(X, A)) + p_{1|0}(X, A) \mid A = a]$$

For each $u \in \mathcal{A}$ we enforce that

$$\sum_{r \in \{0,1\}} \mathbb{E}\left[\pi_r(X)p_{1|r}(X, A) \mid A = u\right] = \sum_{r \in \{0,1\}} \mathbb{E}\left[\pi_r(X, A)p_{1|r}(X, A)\right]$$

We can write this in the generic notation given previously by letting $\mathcal{J} = \mathcal{A} \cup \{\circ\}$,

$$g_j(O, \pi(X); \eta) = \pi_1(X)(p_{1|1}(X, A) - p_{1|0}(X, A)) + p_{1|0}(X, A), \forall j.$$

We let the conditioning events $\mathcal{E}_a = \{A = a\}, \mathcal{E}_\circ = \{\text{True}\}$, i.e. conditioning on the latter is equivalent to evaluating the marginal expectation. Then we express Equation (5) as a set of equality constraints $h_a(\pi) = h_\circ(\pi)$, leading to pairs of inequality constraints,

$$\begin{cases} h_u(\pi) - h_\circ(\pi) \leq 0 \\ h_\circ(\pi) - h_u(\pi) \leq 0 \end{cases}_{u \in \mathcal{A}}$$

The corresponding coefficients of $M$ over this enumeration over groups ($\mathcal{A}$) and epigraphical enforcement of equality ($\{+, -\}$) equation (1), gives $\mathcal{K} = \mathcal{A} \times \{+, -\}$ so that $M_{(a,+),a'} = \mathbf{1}\{a' = a\}, M_{(a,+),\star} = -1, M_{(a,-),a'} = -\mathbf{1}\{a' = a\}, M_{(a,-),\star} = 1$, and $\mathbf{d} = \mathbf{0}$. Further we can relax equality to small amounts of constraint relaxation by instead setting $d_k > 0$ for some (or all) $k$.

Next, we discuss a more complicated fairness measure. We first discuss identification and estimation before we also describe how to incorporate it in the generic framework.

### B.2.2 Responder-dependent fairness measures

We consider a responder framework on outcomes (under our conditional exclusion restriction). Because the contribution to the AEE is indeed from the responder strata, this corresponds to additional estimation of the responder stratum.

We enumerate the four possible realizations of potential outcomes (given any fixed recommendation) as $(Y(0(r)), Y(1(r))) \in \{0,1\}^2$. We call units with $(Y(0(r)), Y(1(r))) = (0,1)$ responders, $(Y(0(r)), Y(1(r))) = (1,0)$ anti-responders, and $Y(0(r)) = Y(1(r))$ non-responders. Such a decomposition is general for the binary setting.

**Assumption 8** (Binary outcomes, treatment).

$$T, Y \in \{0,1\}$$

**Assumption 9** (Monotonicity).

$$Y(T(1)) \geq Y(T(0))$$

Importantly, the conditional exclusion restriction of Assumption 2 implies that responder status is independent of recommendation. Conditional on observables, whether a particular individual is a responder is independent of whether someone decides to treat them when recommended. In this way, we study responder status analogous to its use elsewhere in disparity assessment in algorithmic fairness [23, 28]. Importantly, this assumption implies that the conditioning event (of being a responder) is therefore independent of the policy $\pi$, so it can be handled in the same framework. s

We may consider reducing disparities in resource expenditure given responder status.

We may be interested in the probability of receiving treatment assignment given responder status.

**Example 2** (Fair treatment expenditure given responder status).

$$\mathbb{E}[T(\pi) \mid Y(1(R)) > Y(0(R)), A = a] - \mathbb{E}[T(\pi) \mid Y(1(R)) > Y(0(R)), A = b] \leq \epsilon$$

We can obtain identification via regression adjustment:

**Proposition 7** (Identification of treatment expenditure given responder status). Assume Assumptions 8 and 9.

$$P(T(\pi) = 1 \mid A = a, Y(1(\pi)) > Y(0(\pi))) = \frac{\sum_r \mathbb{E}[\pi_r(X) p_{1|r}(X, A)(\mu_1(X, A) - \mu_0(X, A)) \mid A = a])}{\mathbb{E}[(\mu_1(X, A) - \mu_0(X, A)) \mid A = a]}$$

Therefore this can be expressed in the general framework.

**Example 3** (Writing treatment responder-conditional parity in the general constrained classification framework.). For each $u \in \mathcal{A}$ we enforce that

$$\frac{\sum_r \mathbb{E}[\pi_r(X) p_{1|r}(X, A)(\mu_1(X, A) - \mu_0(X, A)) | A = u])}{\mathbb{E}[(\mu_1(X, A) - \mu_0(X, A)) | A = u]} = \frac{\sum_r \mathbb{E}[\pi_r(X) p_{1|r}(X, A)(\mu_1(X, A) - \mu_0(X, A))])}{\mathbb{E}[(\mu_1(X, A) - \mu_0(X, A))]}$$

We can write this in the generic notation given previously by letting $\mathcal{J} = \mathcal{A} \cup \{\circ\}$,

$$g_j(O, \pi(X); \eta) = \frac{\{\pi_1(X)(p_{1|1}(X, A) - p_{1|0}(X, A)) + p_{1|0}(X, A)\}(\mu_1(X, A) - \mu_0(X, A))}{\mathbb{E}[(\mu_1(X, A) - \mu_0(X, A)) \mid A = a]}, \forall j.$$

Let $\mathcal{E}_a^j = \{A = a_j\}, \mathcal{E}_\circ = \{\text{True}\}$, and we express Equation (5) as a set of equality constraints of the above moment $h_a(\pi) = h_\circ(\pi)$, leading to pairs of inequality constraints,

$$\begin{cases} h_u(\pi) - h_\circ(\pi) \leq 0 \\ h_\circ(\pi) - h_u(\pi) \leq 0 \end{cases}_{u \in \mathcal{A}}$$

The corresponding coefficients of $M$ proceed analogously as for treatment parity.

### B.2.3 Best-response oracles

**Best-responding classifier $\pi$, given $\lambda$: $\text{BEST}_\pi(\lambda)$** The best-response oracle, given a particular $\lambda$ value, optimizes the Lagrangian given $\pi$:

$$L(\pi, \lambda) = \hat{V}(\pi) + \lambda^\top (M\hat{h}(\pi) - \hat{d})$$
$$= \hat{V}(\pi) - \lambda^\top \hat{d} + \sum_{k,j} \frac{M_{k,j} \lambda_k}{p_j} \mathbb{E}_n [g_j(O, \pi) 1\{O \in \mathcal{E}_j\}].$$

**Best-responding Lagrange multiplier $\lambda$, given $\pi$:** $\text{BEST}_{\boldsymbol{\lambda}}(Q)$ is the best response of the $\Lambda$ player. It can be chosen to be either $0$ or put all the mass on the most violated constraint. Let $\gamma(Q) := Mh(Q)$ denote the constraint values, then $\text{BEST}_{\boldsymbol{\lambda}}(Q)$ returns

$$\begin{cases} \mathbf{0} & \text{if } \widehat{\gamma}(Q) \leq \widehat{\mathbf{c}} \\ B\mathbf{e}_{k^*} & \text{otherwise, where } k^* = \arg\max_k \left[ \widehat{\gamma}_k(Q) - \widehat{c}_k \right] \end{cases}$$

### B.2.4 Weighted classification reduction

There is a well-known reduction of optimizing the zero-one loss for policy learning to weighted classification. A cost-sensitive classification problem is

$$\arg\min_{\pi_1} \sum_{i=1}^{n} \pi_1\left(X_i\right) C_i^1 + \left(1 - \pi_1\left(X_i\right)\right) C_i^0$$

The weighted classification error is $\sum_{i=1}^{n} W_i \mathbf{1}\left\{h\left(X_i\right) \neq Y_i\right\}$ which is an equivalent formulation if $W_i = \left|C_i^0 - C_i^1\right|$ and $Y_i = \mathbf{1}\left\{C_i^0 \geq C_i^1\right\}$.

The reduction to weighted classification is particularly helpful since taking the Lagrangian will introduce datapoint-dependent penalties that can be interpreted as additional weights. We can consider the centered regret $J(\pi) = \mathbb{E}[Y(\pi)] - \frac{1}{2}\mathbb{E}[\mathbb{E}[Y \mid R = 1, X] + \mathbb{E}[Y \mid R = 0, X]]$. Then

$$J(\theta) = J(\text{sgn}(g_\theta(\cdot))) = \mathbb{E}[\text{sgn}(g_\theta(X))\{\psi\}]$$

where $\psi$ can be one of, where $\mu_r^R(X) = \mathbb{E}[Y \mid R = r, X]$,

$$\psi_{DM} = (p_{1|1}(X) - p_{1|0}(X))(\mu_1(X) - \mu_0(X)), \psi_{IPW} = \frac{RY}{e_R(X)}, \psi_{DR} = \psi_{DM} + \psi_{IPW} + \frac{R\mu^R(X)}{e_R(X)}$$

We can apply the standard reduction to cost-sensitive classification since $\psi_i \, \text{sgn}(g_\theta(X_i)) = |\psi_i| \, (1 - 2\mathbb{I}\left[\text{sgn}(g_\theta(X_i)) \neq \text{sgn}(\psi_i)\right])$. Then we can use surrogate losses for the zero-one loss,

$$L(\theta) = \mathbb{E}[|\psi| \, \ell(g_\theta(X), \text{sgn}(\psi))]$$

Although many functional forms for $\ell(\cdot)$ are Fisher-consistent, the logistic (cross-entropy) loss will be particularly relevant: $l(g, s) = 2\log(1 + \exp(g)) - (s + 1)g$.

**Example 4** (Treatment parity, continued (weighted classification reduction))**.** The cost-sensitive reduction for a vector of Lagrange multipliers can be deduced by applying the weighted classification reduction to the Lagrangian:

$$L(\beta) = \mathbb{E}\left[|\tilde{\psi}^\lambda| \ell\left(g_\beta(X), \text{sgn}(\tilde{\psi}^\lambda)\right)\right], \qquad \text{where } \tilde{\psi}^\lambda = \psi + \frac{\lambda_A}{p_A}(p_{1|1} - p_{1|0}) - \sum_{a \in \mathcal{A}} \lambda_a.$$

where $p_a := \hat{P}(A = a)$ and $\lambda_a := \lambda_{(a,+)} - \lambda_{(a,-)}$, effectively replacing two non-negative Lagrange multipliers by a single multiplier, which can be either positive or negative.

**Example 5** (Responder-conditional treatment parity, continued)**.** The Lagrangian is $L(\beta) = \mathbb{E}\left[|\tilde{\psi}^\lambda| \ell\left(g_\beta(X), \text{sgn}(\tilde{\psi}^\lambda)\right)\right]$ with weights:

$$\tilde{\psi}^\lambda = \psi + \frac{\lambda_A}{p_A} \frac{(p_{1|1} - p_{1|0})(\mu_1 - \mu_0)}{\mathbb{E}_n[(\mu_1(X, A) - \mu_0(X, A)) \mid A = a]} - \sum_{a \in \mathcal{A}} \lambda_a.$$

where $p_a := \hat{P}(A = a)$ and $\lambda_a := \lambda_{(a,+)} - \lambda_{(a,-)}$.

## B.3 Proofs

*Proof of Proposition 7.*

$$P(T(\pi) = 1 \mid A = a, Y(1(\pi)) > Y(0(\pi)))$$

$$= \frac{P(T(\pi) = 1, Y(1(r)) > Y(0(r)) \mid A = a)}{P(Y(1(\pi)) > Y(0(\pi)) \mid A = a)} \qquad \text{by Bayes' rule}$$

$$= \frac{P(T(\pi) = 1, Y(1) > Y(0) \mid A = a)}{P(Y(1) > Y(0) \mid A = a)} \qquad \text{by Assumption 2}$$

$$= \frac{\sum_r \mathbb{E}[\mathbb{E}[\pi_r(X)\mathbb{I}\left[T(r) = 1\right]\mathbb{I}\left[Y(1) > Y(0)\right] \mid A = a, X]])}{P(Y(1) > Y(0) \mid A = a)} \qquad \text{by iter. exp}$$

$$\frac{\sum_r \mathbb{E}[\pi_r(X)p_{1|r}(X, A)(\mu_1(X, A) - \mu_0(X, A)) \mid A = a])}{\mathbb{E}[(\mu_1(X, A) - \mu_0(X, A)) \mid A = a]} \qquad \text{by Proposition 1}$$

$\square$

## C   Proofs

### C.1   Proofs for generalization under unconstrained policies

**Proposition 8** (Policy value generalization). Assume the nuisance models $\eta = [p_{1|0}, p_{1|1}, \mu_1, \mu_0, e_r(X)]^{\top}, \eta \in H$ are consistent and well-specified with finite VC-dimension $V_{\eta}$ over the product function class $H$. We provide a proof for the general case, including doubly-robust estimators, which applies to the statement of Proposition 8 by taking $\eta = [p_{1|0}, p_{1|1}, \mu_1, \mu_0]$.

Let $\Pi = \{\mathbb{I}\{\mathbb{E}[L(\lambda, X, A; \eta) \mid X] > 0 \colon \lambda \in \mathbb{R}; \eta \in \mathcal{F}\}$.

$$\sup_{\pi \in \Pi, \lambda \in \mathbb{R}} |(\mathbb{E}_n[\pi L(\lambda, X, A)] - \mathbb{E}[\pi L(\lambda, X, A)])| = O_p(n^{-\frac{1}{2}})$$

The generalization bound allows deducing risk bounds on the out-of-sample value:

**Corollary 2.**
$$\mathbb{E}[L(\hat{\lambda}, X, A)_+] \leq \mathbb{E}[L(\lambda^*, X, A)_+] + O_p(n^{-\frac{1}{2}})$$

*Proof of Proposition 8.* We study a general Lagrangian, which takes as input pseudo-outcomes $\psi^{t|r}(O; \eta), \psi^{y|t}(O; \eta), \psi^{1|0, \Delta A}$ where each satisfies that

$$\mathbb{E}[\psi^{t|r}(O; \eta) \mid X, A] = p_{1|1}(X, A) - p_{1|0}(X, A)$$
$$\mathbb{E}[\psi^{y|t}(O; \eta) \mid X, A] = \tau(X, A)$$
$$\mathbb{E}[\psi^{1|0, \Delta A} \mid X] = p_{1|0}(X, a) - p_{1|0}(X, b)$$

We make high-level stability assumptions on pseudooutcomes $\psi$ relative to the nuisance functions $\eta$ (these are satisfied by standard estimators that we will consider):

**Assumption 10.** $\psi^{t|r}, \psi^{y|t}, \psi^{1|0, \Delta A}$ respectively are Lipschitz contractions with respect to $\eta$ and bounded

We study a generalized Lagrangian of an optimization problem that took these pseudooutcome estimates as inputs:

$$L(\lambda, X, A; \eta) = \psi_{t|r}(O; \eta) \left\{ \psi_{y|t}(O; \eta) + \frac{\lambda}{p(A)} (\mathbb{I}[A = a] - \mathbb{I}[A = b]) \right\} + \lambda(\psi^{1|0, \Delta A}(O; \eta))$$

We will show that

$$\sup_{\pi \in \Pi, \lambda \in \mathbb{R}} |(\mathbb{E}_n[\pi L(\lambda, X, A)] - \mathbb{E}[\pi L(\lambda, X, A)])| = O_p(n^{-\frac{1}{2}})$$

which, by applying the generalization bound twice gives that

$$\mathbb{E}_n[\pi L(\lambda, X, A)] = \mathbb{E}[\pi L(\lambda, X, A)]) + O_p(n^{-\frac{1}{2}})$$

Write Lagrangian as

$$\max_{\pi} \min_{\lambda} = \min_{\lambda} \max_{\pi} = \min_{\lambda} \mathbb{E}[L(O, \lambda; \eta)_+]$$

Suppose the Rademacher complexity of $\eta_k$ is given by $\mathcal{R}(H_k)$, so that [7, Thm. 12] gives that the Rademacher complexity of the product nuisance class $H$ is therefore $\sum_k \mathcal{R}(H_k)$. The main result follows by applying vector-valued extensions of Lipschitz contraction of Rademacher complexity given in [36]. Suppose that $\psi^{t|r}, \psi^{y|t}, \psi^{1|0, \Delta A}$ are Lipschitz with constants $C_{t|r}^L, C_{y|t}^L, C_{1|0, \Delta A}^L$.

We establish VC-properties of

$$\mathcal{F}_{L_1}(O_{1:n}) = \{(g_{\eta}(O_1), g_{\eta}(O_i), \dots g_{\eta}(O_n)) \colon \eta \in H\}, \text{ where } g_{\eta}(O) = \psi_{t|r}(O; \eta) \psi_{y|t}(O; \eta)$$

$$\mathcal{F}_{L_2}(O_{1:n}) = \{(h_{\eta}(O_1), h_{\eta}(O_i), \dots h_{\eta}(O_n)) \colon \eta \in H\}, \text{ where } h_{\eta}(O) = \psi_{t|r}(O; \eta) \frac{\lambda}{p(A)} (\mathbb{I}[A = a] - \mathbb{I}[A = b])$$

$$\mathcal{F}_{L_3}(O_{1:n}) = \{(m_{\eta}(O_1), m_{\eta}(O_i), \dots m_{\eta}(O_n)) \colon \eta \in H\}, \text{ where } m_{\eta}(O) = \lambda(\psi^{1|0, \Delta A}(O; \eta))$$

and the function class for the truncated Lagrangian,

$$\mathcal{F}_{L_+} = \{\{(g_\eta(O_i) + h_\eta(O_i) + m_\eta(O_i))_+\}_{1:n} : g \in \mathcal{F}_{L_1}(O_{1:n}), h \in \mathcal{F}_{L_2}(O_{1:n}), m \in \mathcal{F}_{L_3}(O_{1:n}), \eta \in H\}$$

[36, Corollary 4] (and discussion of product function classes) gives the following: Let $\mathcal{X}$ be any set, $(x_1, \ldots, x_n) \in \mathcal{X}^n$, let $F$ be a class of functions $f : \mathcal{X} \to \ell_2$ and let $h_i : \ell_2 \to \mathbb{R}$ have Lipschitz norm $L$. Then

$$\mathbb{E} \sup_{\eta \in H} \sum_i \epsilon_i \psi_i(\eta(O_i)) \leq \sqrt{2} L \mathbb{E} \sup_{\eta \in H} \sum_{i,k} \epsilon_{ik} \eta(O_i) \leq \sqrt{2} L \sum_k \mathbb{E} \sup_{\eta_k \in H_k} \sum_i \epsilon_i \eta_k(O_i) \quad (6)$$

where $\epsilon_{ik}$ is an independent doubly indexed Rademacher sequence and $f_k(x_i)$ is the $k$-th component of $f(x_i)$.

Applying Equation (6) to each of the component classes $\mathcal{F}_{L_1}(O_{1:n}), \mathcal{F}_{L_2}(O_{1:n}), \mathcal{F}_{L_3}(O_{1:n})$, and Lipschitz contraction [7, Thm. 12.4] of the positive part function $\mathcal{F}_{L_+}$, we obtain the bound

$$\sup_{\lambda,\eta} |\mathbb{E}_n[L(O, \lambda; \eta)_+] - \mathbb{E}[L(O, \lambda; \eta)_+]| \leq \sqrt{2}(C_{t|r}^L C_{y|t}^L + C_{t|r}^L B_{p_a} B + B C_{1|0,\Delta A}^L) \sum_k \mathcal{R}(H_k)$$

$\square$

**Proposition 9** (Threshold solutions). Define

$$L(\lambda, X, A) = (p_{1|1}(X, A) - p_{1|0}(X, A)) \left\{ \tau(X, A) + \frac{\lambda}{p(A)} (\mathbb{I}[A = a] - \mathbb{I}[A = b]) \right\} + \lambda(p_{1|0}(X, a) - p_{1|0}(X, b))$$

$$\lambda^* \in \arg\min_\lambda \mathbb{E}[L(\lambda, X, A)_+], \quad \pi^*(x, u) = \mathbb{I}\{L(\lambda^*, X, u) > 0\}$$

If instead $d(x)$ is a function of covariates $x$ only,

$$\lambda^* \in \arg\min_\lambda \mathbb{E}[\mathbb{E}[L(\lambda, X, A) \mid X]_+], \quad \pi^*(x) = \mathbb{I}\{\mathbb{E}[L(\lambda^*, X, A) \mid X] > 0\}$$

*Proof of Proposition 9.* The characterization follows by strong duality in infinite-dimensional linear programming [42]. Strict feasibility can be satisfied by, e.g. solving eq. (3) to set ranges for $\epsilon$. $\square$

## C.2 Proofs for robust characterization

*Proof of Proposition 5.*

$$V(\pi) = \sum_{t\in\mathcal{T}, r\in\{0,1\}} \mathbb{E}[\pi_r(X)\mathbb{E}[c_{rt}(Y(t))\mathbb{I}[T(r)=t] \mid R=r,X]]$$

$$= \sum_{t\in\mathcal{T}, r\in\{0,1\}} \mathbb{E}[\pi_r(X)\mathbb{E}[c_{rt}(Y(t)) \mid R=r,X]P(T(r)=t \mid R=r,X)] \qquad \text{unconf.}$$

$$= \sum_{t\in\mathcal{T}, r\in\{0,1\}} \mathbb{E}[\pi_r(X)\mathbb{E}[c_{rt}(Y(t)) \mid X]P(T(r)=t \mid R=r,X)] \qquad \text{Assumption 2 (ER)}$$

$$\tag{7}$$

$$= \sum_{t\in\mathcal{T}, r\in\{0,1\}} \mathbb{E}\left[\pi_r(X)\mathbb{E}\left[c_{rt}(Y(t))\frac{\mathbb{I}[T(r)=t]}{p_t(X)} \mid X\right]P(T(r)=t \mid R=r,X)\right] \qquad \text{unconf.}$$

$$= \sum_{t\in\mathcal{T}, r\in\{0,1\}} \mathbb{E}\left[\pi_r(X)\left\{\mathbb{E}\left[c_{rt}(Y(t))\frac{\mathbb{I}[T(r)=t]}{p_t(X)} + \left(1 - \frac{T}{p_t(X)}\right)\mu_t(X) \mid X\right]p_{t|r}(X)\right\}\right] \qquad \text{control variate}$$

$$= \sum_{t\in\mathcal{T}, r\in\{0,1\}} \mathbb{E}\left[\pi_r(X)\left\{\left\{c_{rt}(Y(t))\frac{\mathbb{I}[T(r)=t]}{p_t(X)} + \left(1 - \frac{T}{p_t(X)}\right)\mu_t(X)\right\}p_{t|r}(X)\right\}\right] \qquad \text{(LOTE)}$$

where $p_t(X) = P(T=t \mid X)$ (marginally over $R$ in the observational data) and (LOTE) is an abbreviation for the law of total expectation. $\qquad\square$

*Proof of Lemma 1.*

$$\overline{V}_{no}(\pi) := \max_{q_{tr}(X)\in\mathcal{U}}\left\{\sum_{t\in\mathcal{T}, r\in\{0,1\}} \mathbb{E}[\pi_r(X)\mu_t(X)q_{tr}(X)\mathbb{I}[X\in\mathcal{X}^{no}]]]\right\}$$

$$= \max_{q_{tr}(X)\in\mathcal{U}}\left\{\sum_{t\in\mathcal{T}, r\in\{0,1\}} \mathbb{E}[\pi_r(X)\mathbb{E}[Y \mid T=t,X]q_{tr}(X)\mathbb{I}[X\in\mathcal{X}^{no}]]]\right\}$$

Note the objective function can be reparametrized under a surjection of $q_{t|r}(X)$ to its marginalization, i.e. marginal expectation over a $\{T=t\}$ partition (equivalently $\{T=t, A=a\}$ partition for a fairness-constrained setting).

Define
$$\beta_{t|r}(a) := \mathbb{E}[q_{t|r}(X,A) \mid T=t, A=a], \beta_{t|r} := \mathbb{E}[q_{t|r}(X,A) \mid T=t]$$

Therefore we may reparametrize $\overline{V}_{no}(\pi)$ as an optimization over constant coefficients (bounded by B):

$$= \max\left\{\sum_{t\in\mathcal{T}, r\in\{0,1\}} \mathbb{E}[\{c_t\beta_{t|r}\}\pi_r(X)\mathbb{E}[Y \mid T=t,X]\mathbb{I}[X\in\mathcal{X}^{no}]]\colon \underline{B}\le c_1\le\overline{B}, c_0=1-c_1\right\}$$

$$= \max\left\{\sum_{t\in\mathcal{T}, r\in\{0,1\}} \mathbb{E}[\{c_t\beta_{t|r}\}\mathbb{E}[Y\pi_r(X) \mid T=t]\mathbb{I}[X\in\mathcal{X}^{no}]]\colon \underline{B}\le c_1\le\overline{B}, c_0=1-c_1\right\} \qquad \text{LOTE}$$

$$= \sum_{t\in\mathcal{T}, r\in\{0,1\}} \mathbb{E}[c_{rt}^*\beta_{t|r}\mathbb{E}[Y\pi_r(X) \mid T=t]\mathbb{I}[X\in\mathcal{X}^{no}]]$$

where $c_{rt}^* = \begin{cases} \overline{B}\mathbb{I}[t=1] + \underline{B}\mathbb{I}[t=0] & \text{if } \mathbb{E}[Y\pi_r(X) \mid T=t] \ge 0 \\ \overline{B}\mathbb{I}[t=0] + \underline{B}\mathbb{I}[t=1] & \text{if } \mathbb{E}[Y\pi_r(X) \mid T=t] < 0 \end{cases}$

$$\square$$

*Proof of proposition 6.*

$$\max_{\pi} \mathbb{E}[c(\pi, T(\pi), Y(\pi))\mathbb{I}[X \notin \mathcal{X}^{\text{no}}]] + \mathbb{E}[c(\pi, T(\pi), Y(\pi))\mathbb{I}[X \in \mathcal{X}^{\text{no}}]] \tag{8}$$

$$\mathbb{E}[T(\pi)\mathbb{I}[X \notin \mathcal{X}^{\text{no}}] \mid A = a] - \mathbb{E}[T(\pi)\mathbb{I}[X \notin \mathcal{X}^{\text{no}}] \mid A = b] \tag{9}$$

$$+ \mathbb{E}[T(\pi)\mathbb{I}[X \in \mathcal{X}^{\text{no}}] \mid A = a] - \mathbb{E}[T(\pi)\mathbb{I}[X \in \mathcal{X}^{\text{no}}] \mid A = b] \leq \epsilon, \forall q_{r1} \in \mathcal{U} \tag{10}$$

Define

$$g_r(x, u) = (\mu_{r1}(x, u) - \mu_{r0}(x, u))$$

then we can rewrite this further and apply the standard epigraph transformation:

$\max t$

$$t - \int_{x \in \mathcal{X}^{\text{no}}} \sum_{u \in \{a,b\}} \sum_{r \in \{0,1\}} \{g_r(x, u)\pi_r(x, u)f(x, u)\}q_{r1}(x, u)\}dx \leq V_{ov}(\pi) + \mathbb{E}[\mu_0], \forall q_{r1} \in \mathcal{U}$$

$$\int_{x \in \mathcal{X}^{\text{no}}} \{f(x \mid a)(\sum_r \pi_r(x, a)q_{r1}(x, a)) - f(x \mid b)(\sum_r \pi_r(x, b)q_{r1}(x, b))\} + \mathbb{E}[\Delta_{ov}T(\pi)] \leq \epsilon, \forall q_{r1} \in \mathcal{U}$$

Project the uncertainty set onto the direct product of uncertainty sets:

$\max t$

$$t - \int_{x \in \mathcal{X}^{\text{no}}} \sum_{u \in \{a,b\}} \sum_{r \in \{0,1\}} \{g_r(x, u)\pi_r(x, u)f(x, u)\}q_{r1}(x, u)\}dx \leq V_{ov}(\pi) + \mathbb{E}[\mu_0], \forall q_{r1} \in \mathcal{U}_{\infty}$$

$$\int_{x \in \mathcal{X}^{\text{no}}} \{f(x \mid a)(\sum_r \pi_r(x, a)q_{r1}(x, a)) - f(x \mid b)(\sum_r \pi_r(x, b)q_{r1}(x, b))\} + \mathbb{E}[\Delta_{ov}T(\pi)] \leq \epsilon, \forall q_{r1} \in \mathcal{U}_{\in}$$

Clearly robust feasibility of the resource parity constraint over the interval is obtained by the highest/lowest bounds for groups $a, b$, respectively:

$\max t$

$$t - \int_{x \in \mathcal{X}^{\text{no}}} \sum_{u \in \{a,b\}} \sum_{r \in \{0,1\}} \{g_r(x, u)\pi_r(x, u)f(x, u)\}q_{r1}(x, u)\}dx \leq V_{ov}(\pi) + \mathbb{E}[\mu_0], \forall q_{r1} \in \mathcal{U}_{\infty}$$

$$\int_{x \in \mathcal{X}^{\text{no}}} \{f(x \mid a)(\sum_r \pi_r(x, a)\overline{B}_r(x, a)) - f(x \mid b)(\sum_r \pi_r(x, b)\underline{B}_r(x, u))\} + \mathbb{E}[\Delta_{ov}T(\pi)] \leq \epsilon$$

We define

$$\delta_{r1}(x, u) = \frac{2(q_{r1}(x, u) - \underline{B}_r(x, u))}{\overline{B}_r(x, u) - \underline{B}_r(x, u)} - (\overline{B}_r(x, u) - \underline{B}_r(x, u)),$$

then

$$\{\underline{B}_r(x, u) \leq q_{r1}(x, u) \leq \overline{B}_r(x, u)\} \implies \{\|\delta_{r1}(x, u)\|_{\infty} \leq 1\}$$

and

$$q_{r1}(x, u) = \underline{B}_r(x, u) + \frac{1}{2}(\overline{B}_r(x, u) - \underline{B}_r(x, u))(\delta_{r1}(x, u) + 1).$$

For brevity we denote $\Delta B = (\overline{B}_r(x, u) - \underline{B}_r(x, u))$, so

$\max t$

$$t + \max_{\substack{\|\delta_{r1}(x,u)\|_{\infty} \leq 1 \\ r \in \{0,1\}, u \in \{a,b\}}} \left\{ -\int_{x \in \mathcal{X}^{\text{no}}} \sum_{u \in \{a,b\}} \sum_{r \in \{0,1\}} \{g_r(x, u)\pi_r(x, u)f(x, u)\}\frac{1}{2}\Delta B(x, u)\delta_{r1}(x, u)dx \right\} - c_1(\pi) \leq V_{ov}(\pi) + \mathbb{E}$$

$$\int_{x \in \mathcal{X}^{\text{no}}} \{f(x \mid a)(\sum_r \pi_r(x, a)\overline{B}_r(x, a)) - f(x \mid b)(\sum_r \pi_r(x, b)\underline{B}_r(x, u))\} + \mathbb{E}[\Delta_{ov}T(\pi)] \leq \epsilon,$$

where

$$c_1(\pi) = \int_{x \in \mathcal{X}^{\text{no}}} \sum_{u \in \{a,b\}} \sum_{r \in \{0,1\}} \{g_r(x,u)\pi_r(x,u)f(x,u)\}(\underline{B}_r(x,u) + \frac{1}{2}(\overline{B}_r(x,u) - \underline{B}_r(x,u)))dx$$

This is equivalent to:

$$\max t$$

$$t + \int_{x \in \mathcal{X}^{\text{no}}} \sum_{u \in \{a,b\}} \sum_{r \in \{0,1\}} |-g_r(x,u)\pi_r(x,u)f(x,u)| \frac{1}{2}\Delta B(x,u)dx - c_1(\pi) \le V_{ov}(\pi) + \mathbb{E}[\mu_0]$$

$$\int_{x \in \mathcal{X}^{\text{no}}} \{f(x \mid a)(\sum_r \pi_r(x,a)\overline{B}_r(x,a)) - f(x \mid b)(\sum_r \pi_r(x,b)\underline{B}_r(x,u))\} + \mathbb{E}[\Delta_{ov}T(\pi)] \le \epsilon$$

Undoing the epigraph transformation, we obtain:

$$\max V_{ov}(\pi) + \mathbb{E}[\mu_0] + c_1(\pi) - \int_{x \in \mathcal{X}^{\text{no}}} \sum_{u \in \{a,b\}} \sum_{r \in \{0,1\}} |-g_r(x,u)\pi_r(x,u)f(x,u)| \frac{1}{2}\Delta B(x,u)dx$$

$$\int_{x \in \mathcal{X}^{\text{no}}} \{f(x \mid a)(\sum_r \pi_r(x,a)\overline{B}_r(x,a)) - f(x \mid b)(\sum_r \pi_r(x,b)\underline{B}_r(x,u))\} + \mathbb{E}[\Delta_{ov}T(\pi)] \le \epsilon$$

and simplifying the absolute value:

$$\max V_{ov}(\pi) + \mathbb{E}[\mu_0] + c_1(\pi) - \int_{x \in \mathcal{X}^{\text{no}}} \sum_{u \in \{a,b\}} \sum_{r \in \{0,1\}} |g_r(x,u)\pi_r(x,u)f(x,u)| \frac{1}{2}\Delta B(x,u)dx$$

$$\int_{x \in \mathcal{X}^{\text{no}}} \{f(x \mid a)(\sum_r \pi_r(x,a)\overline{B}_r(x,a)) - f(x \mid b)(\sum_r \pi_r(x,b)\underline{B}_r(x,u))\} + \mathbb{E}[\Delta_{ov}T(\pi)] \le \epsilon$$

$$\square$$

### C.3 Proofs for general fairness-constrained policy optimization algorithm and analysis

We begin with some notation that will simplify some statemetns. Define, for observation tuples $O \sim (X, A, R, T, Y)$, the value estimate $v(Q; \eta)$ given some pseudo-outcome $\psi(O; \eta)$ dependent on observation information and nuisance functions $\eta$. (We often suppress notation of $\eta$ for brevity). We let estimators sub/super-scripted by 1 denote estimators from the first dataset.

$$v^{(\cdot)}(Q) = \mathbb{E}_{\pi \sim Q}[\pi \psi_{(\cdot)} \mid O], \text{ for } (\cdot) \in \{\emptyset, \mathrm{DR}\}$$
$$V^{(\cdot)}(Q) = \mathbb{E}[v^{(\cdot)}(Q)]$$
$$\hat{V}_1^{(\cdot)}(Q) = \mathbb{E}_{n_1}[v^{(\cdot)}(Q)]$$
$$g_j(O; Q) = \mathbb{E}_{\pi \sim Q}[g_j(O; \pi) \mid O, \mathcal{E}_j]$$
$$h_j(Q) = \mathbb{E}[g_j(O; Q) \mid \mathcal{E}_j]$$
$$\hat{h}_j^1(Q) = \mathbb{E}_{n_1}[g_j(O; Q) \mid \mathcal{E}_j]$$

#### C.3.1 Preliminaries: results from other works used without proof

**Theorem 3** (Thm. 3, [1] (saddle point generalization bound (non-localized)) ). *Let $\rho :=$ $\max_h \|M\hat{\mu}(h) - \hat{c}\|_\infty$. Let Assumption 1 hold for $C' \geq 2C + 2 + \sqrt{\ln(4/\delta)/2}$, where $\delta > 0$. Let $Q^\star$ minimize $V(Q)$ subject to $M\mu(Q) \leq c$. Then Algorithm 1 with $\nu \propto n^{-\alpha}$, $B \propto n^\alpha$ and $\eta \propto \rho^{-2}n^{-2\alpha}$ terminates in $O\left(\rho^2 n^{4\alpha} \ln |\mathcal{K}|\right)$ iterations and returns $\hat{Q}$. If $np_j^\star \geq 8\log(2/\delta)$ for all $j$, then with probability at least $1 - (|\mathcal{J}| + 1)\delta$ then for all $k$ $\hat{Q}$ satisfies:*

$$V(\hat{Q}) \leq V(Q^\star) + \widetilde{O}\left(n^{-\alpha}\right)$$
$$\gamma_k(\widehat{Q}) \leq c_k + \frac{1 + 2\nu}{B} + \sum_{j \in \mathcal{J}} |M_{k,j}| \widetilde{O}\left((np_j^\star)^{-\alpha}\right)$$

The proof of [1, Thm. 3] is modular in invoking Rademacher complexity bounds on the objective function and constraint moments, so that invoking standard Rademacher complexity bounds for off-policy evaluation/learning [5, 45] yields the above statement for $V(\pi)$ (and analogously, randomized policies by [7, Thm. 12.2] giving stability for convex hulls of policy classes).

**Lemma 2** (Lemma 4, [17])**.** *Consider a function class $\mathcal{F} : \mathcal{X} \to \mathbb{R}^d$, with $\sup_{f \in \mathcal{F}} \|f\|_{\infty,2} \leq 1$ and pick any $f^* \in \mathcal{F}$. Assume that $v(\pi)$ is L-Lipschitz in its first argument with respect to the $\ell_2$ norm and let:*

$$Z_n(r) = \sup_{Q \in \mathcal{Q}} \{|\mathbb{E}_n[\hat{v}(Q) - \hat{v}(Q^*)] - \mathbb{E}[v(Q) - v(Q^*)]| : \mathbb{E}[(\mathbb{E}_{\pi \sim Q}[v(\pi)] - \mathbb{E}_{\pi \sim Q^*}[v(\pi)])^2]^{\frac{1}{2}} \leq r\}$$

*Then for some universal constants $c_1, c_2$ :*

$$\Pr\left[Z_n(r) \geq 16L \sum_{t=1}^d \mathcal{R}\left(r, \mathrm{conv}(\Pi_t) - Q_t^*\right) + u\right] \leq c_1 \exp\left\{-\frac{c_2 nu^2}{L^2 r^2 + 2Lu}\right\}$$

*Moreover, if $\delta_n$ is any solution to the inequalities:*

$$\forall t \in \{1, \ldots, d\} : \mathcal{R}\left(\delta; \mathrm{star}\left(\mathrm{conv}(\Pi_t) - Q_t^*\right)\right) \leq \delta^2$$

*then for each $r \geq \delta_n$ :*

$$P\left(Z_n(r) \geq 16Ldr\delta_n + u\right) \leq c_1 \exp\left\{-\frac{c_2 nu^2}{L^2 r^2 + 2Lu}\right\}$$

**Lemma 3** (Concentration of conditional moments ([1, 47]))**.** *For any $j \in \mathcal{J}$, with probability at least $1 - \delta$, for all $Q$,*

$$\left|\widehat{h}_j(Q) - h_j(Q)\right| \leq 2\mathcal{R}_{n_j}(\mathcal{H}) + \frac{2}{\sqrt{n_j}} + \sqrt{\frac{\ln(2/\delta)}{2n_j}}$$

*If $np_j^\star \geq 8\log(2/\delta)$, then with probability at least $1 - \delta$, for all $Q$,*

$$\left|\widehat{h}_j(Q) - h_j(Q)\right| \leq 2\mathcal{R}_{np_j^\star/2}(\mathcal{H}) + 2\sqrt{\frac{2}{np_j^\star}} + \sqrt{\frac{\ln(4/\delta)}{np_j^\star}}$$

**Lemma 4** (Orthogonality (analogous to [12] (Lemma 8), others)). *Suppose the nuisance estimates satisfy a mean-squared-error bound*

$$\max_l \{\mathbb{E}[(\hat{\eta}_l - \eta_l)^2]\}_{l \in [L]} := \chi_n^2$$

*Then w.p. $1 - \delta$ over the randomness of the policy sample,*

$$V(Q_0) - V(\hat{Q}) \leq O(R_{n,\delta} + \chi_n^2)$$

## C.4 Adapted lemmas

In this subsection we collect results similar to those that have appeared previously, but that require substantial additional argumentation in our specific saddle point setting.

**Lemma 5** (Feasible vs. oracle nuisances in low-variance regret slices ([12], Lemma 9) ). *Consider the setting of Corollary 7. Suppose that the mean squared error of the nuisance estimates is upper bounded w.p. $1 - \delta$ by $h_{n,\delta}^2$ and suppose $h_{n,\delta}^2 \leq \epsilon_n$. Then:*

$$V_2^0 = \sup_{\pi, \pi' \in \Pi_* \left(\epsilon_n + 2h_{n,\delta}^2\right)} \mathrm{Var}\left(v_{DR}^0(x; \pi) - v_{DR}^0(x; \pi')\right)$$

*Then $V_2 \leq V_2^0 + O\left(h_{n,\delta}\right)$.*

## C.5 Proof of Theorem 1

*Proof of Theorem 1.* We first study the meta-algorithm with "oracle" nuisance functions $\eta = \eta_0$.

Define

$$\Pi_2\left(\epsilon_n\right) = \left\{\pi \in \Pi : \mathbb{E}_{n_1}[v(Q; \eta_0) - v(\hat{Q}_1; \eta_0)] \leq \epsilon_n, \mathbb{E}_{n_1}\left[g_j(O; \pi, \eta_0) - g_j\left(O; \hat{\pi}_1, \eta_0\right) \mid \mathcal{E}_j\right] \leq \epsilon_n, j \in \hat{\mathcal{I}}_1\right\}$$

$$\mathcal{Q}_2\left(\epsilon_n\right) = \{Q \in \Delta\left(\Pi_2(\epsilon_n)\right)\}$$

$$\mathcal{Q}^*\left(\epsilon_n\right) = \{Q \in \Delta(\Pi) : \mathbb{E}[(v(Q; \eta_0) - v(Q^*; \eta_0)] \leq \epsilon_n, \mathbb{E}[g_j(O; Q, \eta_0) \mid \mathcal{E}_j] - \mathbb{E}[g_j(O; Q^*, \eta_0) \mid \mathcal{E}_j] \leq \epsilon_n\}$$

In the following, we suppress notational dependence on $\eta_0$.

Note that $\hat{Q}_1 \in \mathcal{Q}_2\left(\epsilon_n\right)$.

Step 1: First we argue that w.p. $1 - \delta/6$, $Q^* \in \mathcal{Q}_2$.

Invoking Theorem 3 on the output of the first stage of the algorithm, yields that with probability $1 - \frac{\delta}{6}$ over the randomness in $\mathcal{D}_1$, by choice of $\epsilon_n = \bar{O}(n^{-\alpha})$),

$$V(\hat{Q}_1) \leq V(Q^*) + \epsilon_n/2$$

$$\gamma_k(\hat{Q}_1) \leq d_k + \sum_{j \in \mathcal{J}} |M_{k,j}| \widetilde{O}\left((np_j^*)^{-\alpha}\right) \leq d_k + \epsilon_n/2 \quad \text{for all } k$$

Further, by Lemma 2,

$$\sup_{Q \in \mathcal{Q}} |\mathbb{E}_{n_1}[(v(Q) - v(Q^*))] - \mathbb{E}[(v(Q) - v(Q^*))]| \leq \epsilon_n/2$$

$$\sup_{Q \in \mathcal{Q}} |\mathbb{E}_{n_1}[(g(O; Q) - g(O; Q^*))] - \mathbb{E}[(g(O; Q) - g(O; Q^*))]| \leq \epsilon_n/2$$

Therefore, with high probability on the good event, $Q^* \in \mathcal{Q}_2$.

Step 2: Again invoking Theorem 3, this time on the output of the second stage of the algorithm with function space $\Pi_2$ (hence implicitly $\mathcal{Q}_2$), and conditioning on the "good event" that $Q^* \in \mathcal{Q}_2$, we obtain the bound that with probability $\geq 1 - \delta/3$ over the randomness of the second sample $\mathcal{D}_2$,

$$V(\hat{Q}_2) \leq V(Q^*) + \epsilon_n/2$$

$$\gamma_k(\hat{Q}_2) \leq \gamma_k(Q^*) + \epsilon_n/2$$

Step 3: empirical small-regret slices relate to population small-regret slices, and variance bounds

We show that if $Q \in \mathcal{Q}_2$, then with high probability $Q \in \mathcal{Q}_2^0$ (defined on small population value- and constraint-regret slices relative to $\hat{Q}_1$ rather than small empirical regret slices)

$$\mathcal{Q}_2^0 = \{Q \in \text{conv}(\Pi) \colon \left| V(Q) - V(\hat{Q}_1) \right| \leq \epsilon_n/2, \mathbb{E}[g_j(O;Q) - g_j(O;\hat{Q}_1)) \mid \mathcal{E}_j] \leq \epsilon_n, \forall j\}$$

so that w.h.p. $\mathcal{Q}_2 \subseteq \mathcal{Q}_2^0$.

Note that for $Q \in \mathcal{Q}$, w.h.p. $1 - \delta/6$ over the first sample, we have that

$$\sup_{Q \in \mathcal{Q}} \left| \mathbb{E}_n[v(Q) - v(\hat{Q}_1)] - \mathbb{E}[v(Q) - v(\hat{Q}_1)] \right| \leq 2 \sup_{Q \in \mathcal{Q}} |\mathbb{E}_n[v(Q)] - \mathbb{E}[v(Q)]| \leq \epsilon,$$

$$\sup_{Q \in \mathcal{Q}} \left| \mathbb{E}_{n_1}[g_j(O;Q) - g_j(O;\hat{Q}_1) \mid \mathcal{E}_j] - \mathbb{E}[g_j(O;Q) - g_j(O;\hat{Q}_1) \mid \mathcal{E}_j] \right|$$

$$\leq 2 \sup_{Q \in \mathcal{Q}} |\mathbb{E}_{n_1}[g_j(O;Q) \mid \mathcal{E}_j] - \mathbb{E}[g_j(O;Q) \mid \mathcal{E}_j]| \leq \epsilon, \forall j$$

The second bound follows from [7, Theorem 12.2] (equivalence of Rademacher complexity over convex hull of the policy class) and linearity of the policy value and constraint estimators in $\pi$, and hence $Q$.

On the other hand since $Q_1$ achieves low policy regret, the triangle inequality implies that we can contain the true policy by increasing the error radius. That is, for all $Q \in \mathcal{Q}_2$, with high probability $\geq 1 - \delta/3$:

$$|\mathbb{E}[(v(Q) - v(Q^*))]| \leq \left| \mathbb{E}[(v(Q) - v(\hat{Q}_1))] \right| + \left| \mathbb{E}[(v(\hat{Q}_1) - v(Q^*))] \right| \leq \epsilon_n$$

$$|\mathbb{E}[g_j(O;Q) - g_j(O;Q^*) \mid \mathcal{E}_j]| \leq \left| \mathbb{E}[g_j(O;Q) - g_j(O;\hat{Q}_1) \mid \mathcal{E}_j] \right| + \left| \mathbb{E}[g_j(O;\hat{Q}_1) - g_j(O;Q^*) \mid \mathcal{E}_j] \right| \leq \epsilon_n$$

Define the space of distributions over policies that achieve value and constraint regret in the population of at most $\epsilon_n$ :

$$\mathcal{Q}_*(\epsilon_n) = \{Q \in \mathcal{Q} \colon V(Q) - V(Q^*) \leq \epsilon_n, \ \mathbb{E}[g_j(O;Q) - g_j(O;Q^*) \mid \mathcal{E}_j] \leq \epsilon_n, \forall j\},$$

so that on that high-probability event,

$$\mathcal{Q}_2^0(\epsilon_n) \subseteq \mathcal{Q}_*(\epsilon_n). \tag{11}$$

Then on that event with probability $\geq 1 - \delta/3$,

$$r_2^2 = \sup_{Q \in \mathcal{Q}_2} \mathbb{E}[(v(Q) - v(Q^*))^2] \leq \sup_{Q \in \mathcal{Q}^*(\epsilon_n)} \mathbb{E}[(v(Q) - v(Q^*))^2]$$

$$= \sup_{Q \in \mathcal{Q}^*(\epsilon_n)} \text{Var}(v(Q) - v(Q^*)) + \mathbb{E}[(v(Q) - v(Q^*))]^2$$

$$\leq \sup_{Q \in \mathcal{Q}^*(\epsilon_n)} \text{Var}(v(Q) - v(Q^*)) + \epsilon_n^2$$

Therefore:

$$r_2 \leq \sqrt{\sup_{Q \in \mathcal{Q}_*(\epsilon_n)} \text{Var}(v(Q) - v(Q^*))} + 2\epsilon_n = \sqrt{V_2} + 2\epsilon_n$$

Combining this with the local Rademacher complexity bound, we obtain that:

$$\mathbb{E}[v(\hat{Q}_2) - v(Q^*)] = O\left(\kappa\left(\sqrt{V_2} + 2\epsilon_n, \mathcal{Q}_*(\epsilon_n)\right) + \sqrt{\frac{V_2 \log(3/\delta)}{n}}\right)$$

These same arguments apply for the variance of the constraints

$$V_2^j = \sup\{\text{Var}(g_j(O;Q) - g_j(O;Q')) : Q, Q' \in \mathcal{Q}_*(\tilde{\epsilon}_n)\}$$

$\square$

## C.6 Proofs of auxiliary/adapted lemmas

*Proof of Lemma 5.* The proof is analogous to that of [12, Lemma 9] except for the step of establishing that $\pi_* \in \mathcal{Q}^0_{\epsilon_n + O(\chi^2_{n,\delta})}$: in our case we must establish relationships between saddlepoints under estimated vs. true nuisances. We show an analogous version below.

Define the saddle points to the following problems (with estimated vs. true nuisances):

$$(Q^*_{0,0}, \lambda^*_{0,0}) \in \arg\min_Q \max_\lambda \mathbb{E}[v_{DR}(Q; \eta_0)] + \lambda^\top(\gamma_{DR}(Q; \eta_0) - d) \coloneqq L(Q, \lambda; \eta_0, \eta_0) \coloneqq L(Q, \lambda),$$

$$(Q^*_{\eta,0}, \lambda^*_{\eta,0}) \in \arg\min_Q \max_\lambda \mathbb{E}[v_{DR}(Q; \eta)] + \lambda^\top(\gamma_{DR}(Q; \eta_0) - d),$$

$$(Q^*, \lambda^*) \in \arg\min_Q \max_\lambda \mathbb{E}[v_{DR}(Q; \eta)] + \lambda^\top(\gamma_{DR}(Q; \eta) - d).$$

We have that:

$$
\begin{aligned}
\mathbb{E}[v_{DR}(Q^*)] &\leq L(Q^*, \lambda^*; \eta, \eta) + \nu \\
&\leq L(Q^*, \lambda^*; \eta, \eta_0) + \nu + \chi^2_{n,\delta} \\
&\leq L(Q^*, \lambda^*; \eta, \eta_0) + \nu + \chi^2_{n,\delta} && \text{by Lemma 4} \\
&\leq L(Q^*, \lambda^*_{\eta,0}; \eta, \eta_0) + \nu + \chi^2_{n,\delta} && \text{by saddlepoint prop.} \\
&\leq L(Q^*_{\eta,0}, \lambda^*_{\eta,0}; \eta, \eta_0) + \left| L(Q^*_{\eta,0}, \lambda^*_{\eta,0}; \eta, \eta_0) - L(Q^*, \lambda^*_{\eta,0}; \eta, \eta_0) \right| + \nu + \chi^2_{n,\delta} && \text{triangle ineq.} \\
&\leq L(Q^*_{\eta,0}, \lambda^*_{\eta,0}; \eta, \eta_0) + \epsilon_n + \nu + \chi^2_{n,\delta} && \text{assuming } \epsilon_n \geq \chi^2_{n,\delta} \\
&\leq \mathbb{E}[v_{DR}(Q^*_{\eta,0}; \eta)] + \epsilon_n + 2\nu + \chi^2_{n,\delta} && \text{apx. complementary slackness} \\
&\leq \mathbb{E}[v_{DR}(Q^*_{0,0}; \eta)] + \epsilon_n + 2\nu + \chi^2_{n,\delta} && \text{suboptimality}
\end{aligned}
$$

Hence

$$\mathbb{E}[v_{DR}(Q^*; \eta)] - \mathbb{E}[v_{DR}(Q^*_{0,0}; \eta)] \leq \epsilon_n + 2\nu + \chi^2_{n,\delta}.$$

We generally assume that the saddlepoint suboptimality $\nu$ is of lower order than $\epsilon_n$ (since it is under our computational control).

Applying Lemma 4 gives;

$$V(Q^*) - V(Q^*_{0,0}) \leq \epsilon_n + 2\nu + 2\chi^2_{n,\delta}.$$

Define policy classes with respect to small-population regret slices (with a nuisance-estimation enlarged radius):

$$\mathcal{Q}^0(\epsilon) = \{Q \in \Delta(\Pi) \colon V(Q^*_0) - V(Q) \leq \epsilon, \gamma(Q^*_0) - \gamma(Q) \leq \epsilon\}$$

Then we have that

$$V_2^{obj} \leq \sup_{Q \in \mathcal{Q}^0(\epsilon_n)} \text{Var}(v_{DR}(O; \pi) - v_{DR}(O; \pi^*)),$$

where we have shown that $\pi^* \in \mathcal{Q}^0(\epsilon + 2\nu + 2\chi^2_{n,\delta})$.

Following the result of the argumentation in [12, Lemma 9] from here on out gives the result. $\qquad\square$

# D  Case Studies

## D.1  Oregon Health Insurance Study

The Oregon Health Insurance Study [16] is an important study on the causal effect of expanding public health insurance on healthcare utilization, outcomes, and other outcomes. It is based on a randomized controlled trial made possible by resource limitations, which enabled the use of a randomized lottery to expand Medicaid eligibility for low-income uninsured adults. Outcomes of interest included health care utilization, financial hardship, health, and labor market outcomes and political participation.

Because the Oregon Health Insurance Study expanded access to *enroll* in Medicaid, a social safety net program, the effective treatment policy is in the space of *encouragement* to enroll in insurance (via access to Medicaid) rather than direct enrollment. This encouragement structure is shared by many other interventions in social services that may invest in nudges to individuals to enroll, tailored assistance, outreach, etc., but typically do not automatically enroll or automatically initiate transfers. Indeed this so-called *administrative burden* of requiring eligible individuals to undergo a costly enrollment process, rather than automatically enrolling all eligible individuals, is a common policy design lever in social safety net programs. Therefore we expect many beneficial interventions in this consequential domain to have this encouragement structure.

We preprocess the data by partially running the Stata replication file, obtaining a processed data file as input, and then selecting a subset of covariates. These covariates include household information that affected stratified lottery probabilities, socioeconomic demographics, medical status and other health information.

In the notation of our framework, the setup of the optimal/fair encouragement policy design question is as follows:

- $X$ covariates (baseline household information, socioeconomic demographics, health information)

- $A$ race (non-white/white), or gender (female/male)

  These protected attributes were binarized.

- $R$ encouragement: lottery status of expanded eligibility (i.e. invitation to enroll when individual was previously ineligible to enroll)

- $T$: whether the individual is enrolled in insurance ever

  Note that for $R = 1$ this can be either Medicaid or private insurance while for $R = 0$ this is still well-defined as this can be private insurance.

- $Y$: number of doctor visits

  This outcome was used as a measure of healthcare utilization. Overall, the study found statistically significant effects on healthcare utilization. An implicit assumption is that increased healthcare utilization leads to better health outcomes.

We subsetted the data to include complete cases only (i.e. without missing covariates). We learned propensity and treatment propensity models via logistic regression for each group, and used gradient-boosted regression for the outcome model. We first include results for regression adjustment identification.

In Figure 3 we plot descriptive statistics. We include histograms of the treatment responsivity lifts $p_{1|1a}(x, a) - p_{1|0a}(x, a)$. We see some differences in distributions of responsivity by gender and race. We then regress treatment responsivity on the outcome-model estimate of $\tau$. We find substantially more heterogeneity in treatment responsivity by race than by gender: whites are substantially more likely to take up insurance when made eligible, conditional on the same expected treatment effect heterogeneity in increase in healthcare utilization. (This is broadly consistent with health policy discussions regarding mistrust of the healthcare system).

Next we consider imposing treatment parity constraints on an unconstrained optimal policy (defined on these estimates). In Figure 4 we display the objective value, and $\mathbb{E}[T(\pi) \mid A = a]$, for gender and race, respectively, enumerated over values of the constraint. We use costs of 2 for the number of doctors visits and 1 for enrollment in Medicaid (so $\mathbb{E}[T(\pi) \mid A = a]$ is on the scale of probability of

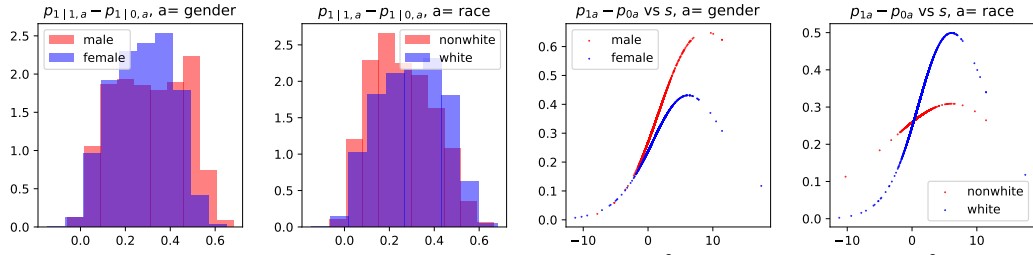

Figure 3: Distribution of lift in treatment probabilities $p_{1|1,a} - p_{1|0,a} = P(T = 1 \mid R = 1, A = a, X) - P(T = 1 \mid R = 0, A = a, X)$, and plot of $p_{1|1,a} - p_{1|0,a}$ vs. $\tau$.

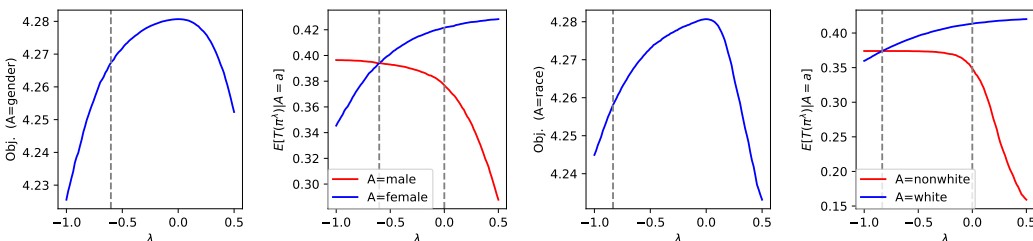

Figure 4: Policy value $V(\pi^\lambda)$, treatment value $\mathbb{E}[T(\pi^\lambda) \mid A = a]$, for $A$ = race, gender.

enrollment). These costs were chosen arbitrarily. Finding optimal policies that improve disparities in group-conditional access can be done with relatively little impact to the overall objective value. These group-conditional access disparities can be reduced from 4 percentage points $(0.04)$ for gender and about 6 percentage points $(0.06)$ for race at a cost of $0.01$ or $0.02$ in objective value (twice the number of doctors' visits). On the other hand, relative improvements/compromises in access value for the "advantaged group" show different tradeoffs. Plotting the tradeoff curve for race shows that, consistent with the large differences in treatment responsivity we see for whites, improving access for blacks. Looking at this disparity curve given $\lambda$ however, we can also see that small values of $\lambda$ can have relatively large improvements in access for blacks before these improvements saturate, and larger $\lambda$ values lead to smaller increases in access for blacks vs. larger decreases in access for whites.

