# OpenReview forum: "Optimal and Fair Encouragement Policy Evaluation and Learning"
_NeurIPS.cc/2023/Conference — NeurIPS 2023 poster_

### Official Review · Reviewer_xiA7 · 2023-07-05

**Soundness:** 3 good
**Presentation:** 4 excellent
**Contribution:** 4 excellent
**Rating:** 7
**Confidence:** 1

**Summary:**

There exist (many) healthcare programs where it is impossible to compel individuals to take treatment; rather, the problem of solving for an optimal policy takes the form of providing beneficial services and recommendations to patients. This paper formalizes a causal framework for these cases and develops statistically improved estimators and robustness checks for the setting of algorithmic recommendations with sufficiently randomized decisions. The contributions of this paper are both theoretical and empirical in nature.

**Strengths:**

I believe that this paper is clearly written. While I struggled to understand the work in this paper, I believe that my challenges were due to not being an expert in this field. In fact, I have learned something about causality from reading this paper.

The problem that this paper is attempting to address is very interesting to me, and the empirical results on a variety of applications illustrated the use and significance of this paper’s contributions.


**Weaknesses:**

I have not identified any core weaknesses in this paper.

There’s an unattributed quote block on line 33.


**Questions:**

N/A

**Limitations:**

Yes.

---

> ### Author Response · Authors · 2023-08-14
>
> Thanks for the review! The quote block is our own emphasis.

---

### Official Review · Reviewer_h3Xk · 2023-07-06

**Soundness:** 3 good
**Presentation:** 3 good
**Contribution:** 3 good
**Rating:** 6
**Confidence:** 3

**Summary:**

The authors study the problem of providing treatment recommendations in systems where those receiving recommendations posses the ability to dissent or deviant from the suggested course of action. Within these types of problems the authors develop a method for providing treatment recommendations which can both account for this ability in recomendees and adhere to notions of fairness in terms of successful treatment adoption. The authors provide several theoretical results characterizing the problem space and their approach as well as experimentally demonstrating its efficacy.  On top of this the authors adapt the classic two-player cost-sensitive learning optimization technique found in prior works to their problem and introduce several interesting additions to account for the additional considerations of their problem.

**Strengths:**


- The problem studied in the paper is important and helps bridge the gap between works on treatment recommendation which ignore autonomy of the recomendee and the real-world in which humans will inevitably express dissatisfaction or apathy towards recommendations.

- The paper is well written and the authors clearly outline their approach, relevant background, and possible limitations.

- The authors provide a mix of theoretical results (characterizing both the problem and priorities of their approach) and experimental results.

- The assumptions made in the authors model are reasonable. The least standard assumption appearing to be Assumption 2, which may not always hold in practice (as the authors point out), but this assumptions is a good jumping-off point for the type of problem the authors wish to investigate. Moreover, in most “reasonable” domains I would expect that the majority of the effect of the recommendation is its ability to alter treatment adoption rates.

- The modifications to the traditional reductions approach are interesting and provide some useful new ideas from a technical standpoint.


**Weaknesses:**

- The experimental results are missing baseline comparisons. For example, what would happen if we assumed a 100% compliance rate in terms of recommendations; applying such an approach would provide information on how much we gain by considering agent autonomy.

- It would be informative to see some type of running time analysis or discussion on how easy it is to solve the objectives presented on line 219. For example, computing a best response amounts to cost sensitive learning at each round, which I would expect is extremely costly unless only a few epochs of cost sensitive learning are performed at each round. Although this approach is heavily inspired by (and similar to) the method in Agarwal et al., which is very fast in practice, it is hard to see how much extra compute is required as a result of the difference between the two approaches. The authors approach feels more similar to adversarial training which is known to be a quite expensive and scale poorly with larger models and feature spaces (but perhaps I am wrong about it). Any clarification would be appreciated!


**Questions:**

See weaknesses

**Limitations:**

See weakensses

---

> ### Author Rebuttal · Authors · 2023-08-08
>
>  Thanks for the summary and assessment of strengths!
>
> We respond to questions/limitations/weaknesses.
>
> 1. Baselines:
>
> Good question. Note that we actually plot the full curve of objective value and constraint reduction in Fig. 2. To your question about what happens if we wrongly assume a 100% compliance rate: optimizing over an unconstrained policy class, the policy decision is the same, assuming monotonicity (that recommending treatment only makes treatment more, not less, likely). Therefore, assuming 100% compliance is equivalent to an unconstrained policy (i.e. \lambda = 0). Although we increase the objective function, we incur great disparities in treatment across groups, because we wrongly assumed 100% compliance.
>
> However, the optimal policy value could indeed be different when we optimize over a *constrained* policy class, even without any _fairness_ constraints. It depends on the magnitude of the “compliance weights” in addition to $\tau=\mu_1(x)-\mu_0(x)$ and the exact functional form. For example, consider a simple two-dimensional case with linear decision rule. Suppose for $x_1 > 0, \tau = -1, x_1<0, \tau=0$ (good reduction in costs when $x_1>0$) but compliance is different: for $ \{x_1>0, x_2 > 0\}$, we have that $p_{1\mid 1}-p_{1\mid 0} = -\epsilon$ but for the other subregion, $ \{x_1>0 , x_2 < 0\}$, we have that $p_{1\mid 1}-p_{1\mid 0} = 1$. Then the optimal encouragement-aware linear decision rule treats  $ \{x_1>0 , x_2 < 0\}$, but wrongly assuming 100% compliance treats $x_1>0$ which does worse! We will add this simple example to illustrate what can go wrong.
> 2. Running-time analysis or discussion:
> Good point; we will add this discussion in the appendix.
> The approach we suggest only has twice the computational complexity of the method of Agarwal et al. because our two-stage variance reduced approach essentially runs their method twice. That’s because policy learning can be reduced to cost-sensitive learning, which is the key computational oracle used in Agarwal et al. That also means that improvements in cost-sensitive learning, such as using stochastic gradient techniques can equivalently be used here and reduce computational complexity. Computationally, one iteration of the approach can be as difficult as solving a mixed-integer program (aka solvable in practice but without poly-time guarantees), or can be solved computationally efficiently with stochastic gradient descent on a surrogate-loss relaxation; both of these are common approaches in practice.

---

> > ### Comment · Reviewer_h3Xk · 2023-08-18
> > **Response to authors**
> >
> > Thank you for the clarifications. My concerns have been addressed. Mentioning the complexity and runtime with respect to methods such as that of Agarwal et al. would help contextualize the efficiency of the method. Based on my reading of the paper, I expected the method to be quite intractable at scale, but based on the rebuttal this is not the case.
> >
> > After reading the the other reviews and the authors' responses, I will maintain my original score.

---

### Official Review · Reviewer_Cr9c · 2023-07-16

**Soundness:** 2 fair
**Presentation:** 1 poor
**Contribution:** 1 poor
**Rating:** 3
**Confidence:** 3

**Summary:**

The paper addresses consequential decision making situation where we have a decision-making policy that outputs decisions (referred to as recommendatinos) R, individuals who may adhere to the decisions and realize a treatment T, which then leads to an outcome Y.  The problem arises, when individuals do not adhere to the decisions (i.e., follow the recommendations) and thus do not realize a treatment T, thus not realize any outcome Y. The decision making policy does not have direct control over treatment, but can only provide decisions (recommendations for treatment). Fainress comes into play, when amongst those poeple that received a positive decision (recommendation) certain social groups are less likely to adhere to the decision and thus realize an outcome. The paper suggests a method to learn a decision making policy that assigns recommendations to individuals as to fulfill certain fairness criteria in the treatment realized, e.g., reducing the disparity in treatment. The authors present a case study on "failure to appear before court" data (PSA-DMF).




**Strengths:**

The paper studies an interesting problem, that can be relevant to the ML community working on consequential decision making). In the classical loan example, we generally assume that an individual decided to be granted a loan will utilize it, overlooking the possibility that some individuals might choose not to realize the loan. When individuals from certain demographics systematically underutilize positive decisions, this has fairness implications that are important to be considered, when designing decision-making policies.

**Weaknesses:**

While I believe the problem tackled is interesting, I strongly believe the paper is not yet ready for publication. The paper lacks structure and clarity. I am enlisting some important points below.

On structure:
1. Section 1: The last paragraph of the introduction summarizes the contribution. It would help, if the authors could point to the specific sections, where these contributions are made and provide an overview of the structure of the paper.
2. Section 3: There is no background section. The paper would benefit from a background section to understand the introduced method better. For example, I do not find the Neyman-Rubin potnetial outcomes framework introduced that the work claims to use (ln. 95-96). There is also no introduction to policy value functions, doubly robust optimization/policies.
3. Section 3: The introduction of notation and concepts may benefit from introducing concepts as definitions, e.g., the cost function. There is also a lack of introduction of notation, e.g., what $p_{t|r}$ refers to.
4. Section 3: The mathematical definitions / equations do not carry any equation numbers. The work could benefit from adding equation numbers, such that certain formulas or definitions are easier accessible.
5. Section 4: This section could benefit from an introduction, what this section is about. I am not understanding, why this is called "Method" and then section 5 also introduces a method ("We now introduce general methodology to handle ...", ln 214)
6. Finally, there is no discussion/limitations/outlook section. The paper would benefit from a section that summarizes the work, discusses its contribution and limiations and provides an outlook.

Clairty: I find it, in general, very difficult to follow the authors in their main idea. I am sharing a selection of difficulties I had with respect to clarity:
1. Section 1: I am missing clarity in the pipeline of decisions, why is human in the loop that makes the final prescription a problem? I understood the problem is that people, even if they receive a prescription, may not comply with it. So I understood the problem to be rather on the subject side that receives the treatment. Could you clarify?
2. Section 2: The related work section cites work relevant to the field. However, I am missing clarity in how the previous work relates to the current work. There is a list of work and description of it, but often I am missing how this relates. For example, work [37] is cited twice in line 62 and then again in line 65, I am not understanding that split. In line 72 there is also a mentioning of "supervised release", without explaining it. You aslo write ln. 81 "a different line of work studies counterfactual risk assesment, which models a different concern", how is that different?
3. Section 3: There is an introduction of social groups "regarding fairness, we will be concerned about disparities in utiltiy and treatment benefits across different groups", but then I find "utility" and "treatment benefits" not explained.
3. Section 3: While assumption 2, 4, 6 are addressed in lines 126 ff. I am missing an explanation of the rest of the assumptions.
4. Section 4 & 5: I am failing to follow the story and propositions. The paper would benefit from a clear explanation in words of i) what the proposition says / what the formulas express and ii) why it is important for the remainer/goal of the paper.
5. Section 6: I am not understanding the motivation or story of the case study in the experimental evaluation. I am missing a description of the experimental setup.
7. Section 6: The figures are missing an explanation what the different parameters are, e.g., $\tau$ etc.
6. A general point, I find the usage of the word "recommendation" difficult in the context of fair decision making (and here the authors use binary decisions as far as I understand). This is because there is a different field of fairness literature that concerns recommendations in teh context of ranking and that is different from decision making.

In addition I have the following comments/concerns about citations:
1. I am missing citations in the introduction. For example, in line 38ff. "a common strategy is to conduct an intention-to-treat analysis"; in line 47 "previous work in algorithmic accountability primarily focuses on auditing recommendations".
2. I looked up citations [20, 21] and did not found "the well-understood notion of non-compliance/non-adherence". Can you point me to this one? Also I found [21] to be a presentation on causal inference, not a (published) paper or book, is this correct? I am not sure about the quality of that citation.

About formatting:
1. line 33 ff. is differently formated than the text below, I am not understanding why.
2. Margin violations, at proposition 2, 5, and 6

Typos:
* ln. 85 "don't" -> do not
* ln. 108 do you mean $\mathcal{T} = \{0,1\}$ instead of $\mathcal{T} \in = \{0,1\}$
* ln. 129 a blank to much after X
* ln. 291 "arbitrarily"

**Questions:**

There might be parts of their methodology or approach I simply did not understand due to the lack of explanation.

1. Section 3: I find the equations in line 112 not introduced, what is $e_r$, what is $\mu_{r, t}$ and what do they mean, what will we need them for?
2. Section 4: line 176 "since algorithmic recommendations are deterministic functions of covariates" - what do you mean by this? Decision policies can be probabilistic [1, 2], as far as I am concerned. Or am I missing the point?
3. Why are you using doubly robust estimation? From the text, I am not understanding why this is necessary. Also it may be worth stating, why you do not simply use inverse propensity scoring.
4. In Section 4 and 5 I am failing to undestand the fairness understanding present in the work. Could you point me to where this is defined and detailled?
5. What are you assumptions in the experimental section as to how individuals realize treatments?


[1] Bechavod, Yahav, et al. "Equal opportunity in online classification with partial feedback." Advances in Neural Information Processing Systems 32 (2019).
[2] Kilbertus, Niki, et al. "Fair decisions despite imperfect predictions." International Conference on Artificial Intelligence and Statistics. PMLR, 2020.




**Limitations:**

A discussion or outlook section is missing. The authors do address limitations of some of their assumptions in lines 126

---

> ### Author Rebuttal · Authors · 2023-08-08
>
> We respond point-by-point below. Note many of these points are *already* in the paper.
>
> 1. Adding section numbers to our narrative description is straightforward and we will do so.
> 2. No, our background is split between the related work and problem setup.
> We introduce policy value functions in lines 113-118. Re: assumptions, see response to Reviewer QSPP, weakness 2. We will go into greater description in the appendix, and include the following sentences about the standard DR estimator:
> > “For complete background we describe estimation improvements in standard causal inference in estimating the average treatment effect. The celebrated doubly-robust estimator for the ATE is [Robins and Rotnitzky 1995]: $$V_{DR}(\pi) = \sum_{t \in \mathcal{T}} E[ \pi(t\mid X)( \mu_t(X) +  \mathbb{I}[T=t] (Y-\mu_t(X)) / e_t(X))  ] $$ Doubly robust estimators use both the outcome model and propensity score model and enjoy 1) two chances at model misspecification: only one of the propensity score and outcome model need to be well-specified and 2) rate double-robustness: we only require the product of the MSE convergence rates to achieve $n^{-\frac 12}$ consistency. Finally, one last perspective on doubly-robust estimators is that they are control variate estimators. This last perspective is closest to the developments in our paper. Looking at the form of the doubly-robust estimator, it is the sum of a zero-mean term to the regression adjustment estimator. “
> 3. Sec. 3 is a succinct formal problem setup. We will add,
> > “The cost function puts recommendations, treatments, and causal outcomes in the same “currency”, for example, a common cost basis to compare utility of, for example, the final purchase value if someone uses a 10 dollar off coupon vs. the cost of someone using treatment, e.g. the 10 dollar discount.”
> 4. Adding equation numbers is straightforward. But we *already* have assumption/proposition labels, e.g. line 112, line 173, 177, 247, 255, 292.
> 5. No, we *already* have an introduction there in lines 146-153: we summarize that we study two regimes, causal identification and estimation. We will add a summary sentence.
> 6. We discuss limitations throughout. See response to 2WiP.
>
> Clarity:
> 1. Algorithmic auditing looks at disparities in $P(R=1|A=1)-P(R=1|A=0)$ instead of $P(T=1|A=1)-P(T=1|A=0)$. This can be an issue whether it is the subject not complying with the treatment, or another decision-maker. We describe this in lines 46-48 (fairness constraints should be on realized treatments, not algorithmic recommendations) and lines 118-120 (the optimal decision rule can be different).
> 2.  No, after most works mentioned, we exactly describe how our work is similar and different. See line 65, 66-68; where we introduce a related work, we contrast with “but (without/not), however, rather than, our focus is instead on”. Counterfactual risk assessment still assesses fairness of the high-risk/low-risk labels (recommendations), not the realized decisions.
> 3. We provided multiple examples of utility of outcomes vs. benefits from treatment access alone in the introduction in 25-31, 36-49.
> 4. The other assumptions are standard in causal inference, as we mentioned. See response to Structure 2).
> 5. We already did this. Every proposition has a sentence right before it describing the result in words: “we discuss causal identification (Prop. 1), we characterize a threshold solution.. (Prop. 2)”. E.g. lines 163-164 say that we use Prop. 2 to obtain the the next proposition. We can make these transitions longer.
> 6. Lines 261-287 and lines 293-300 (interpretation of fairness/treatment costs) describe the story (summarized for space constraints). We will add a few sentences:
> > Judges can choose to detain, (unconditionally) release, or release with supervision (supervised release) defendants when they are arrested before their trial. … lines 262 - lines 267 … Supervised release is an example of the second regime: recommendations are made with a human in the loop who makes the final decision, and we are concerned about disparities in outcomes and treatments.
> 7. See line 285. $\tau = \mu_1(x) - \mu_0(x)$. Will add.
> 8. We use the terms “algorithmic recommendation, recommendation for treatment, encouragement/recommendation” to avoid confusion.
>
> Citations:
> 1. Many papers audit fair binary classification in a consequential domain such as social services (with caseworkers) [1], healthcare [2], criminal justice [3] where decisions are not automated.
> [1] Chouldechova, Alexandra, et al. "A case study of algorithm-assisted decision making in child maltreatment hotline screening decisions." FAT*, 2018.
> [2] Pfohl, Stephen R., Agata Foryciarz, and Nigam H. Shah. "An empirical characterization of fair machine learning for clinical risk prediction." Journal of biomedical informatics 2021.
> [3] Stevenson, Megan T., and Jennifer L. Doleac. "Algorithmic risk assessment in the hands of humans." (2022).
> 2. No, [21] is a book: Hernán MA, Robins JM (2020). Causal Inference: What If. Boca Raton: Chapman & Hall/CRC. See 22.1.
> Pages 12-15 of [20] discuss encouragement designs related to Appendix D.1.
>
> Questions:
> 1. No, we said what they are on line 112: recommendation ($e_r$), treatment propensity ($p_{t\mid r}$) and outcome ($u_{t}$) models for causal identification and estimation.
> 2. line 175 says these are from deterministic binary classifiers (as common in the fairness literature).
> 3. Doubly robust estimators are standard important estimators in causal inference that reduce variance in estimation. See our response to your structure 2.
> 4. Equation (1) has the main fairness constraint we use as an example in the main text. After line 218 we refer to more detail in appendix B.2.1 and B.2.2. We described what fairness constraints are relevant in lines 36-29.
> 5. We said this in lines 272-275 but can add numbers. The ones relevant to treatment realizations are Assumptions 3 (unconfoundedness), 4 (responsivity) and 6 (overlap).

---

> > ### Author Response · Authors · 2023-08-16
> >
> > Thank you again for your concerns. We want to follow up on the rebuttal. We understand that you may have a busy schedule, but we would greatly appreciate it if you could take a moment to review our response and provide any additional feedback.
> >
> > If you find our response useful, please consider updating your score? We hope we have thoroughly clarified that many of these points are in the manuscript, and are absolutely not fundamental flaws, though we will be absolutely sure to be absolutely clear about these.
> >
> > Lastly, if there are any specific areas of concern that we can address or provide additional clarification on, please do not hesitate to let us know.

---

> > > ### Comment · Reviewer_Cr9c · 2023-08-17
> > >
> > > Dear authors,
> > >
> > > Thank you for the comprehensive responses, which have indeed clarified some of the uncertainties I had, for example with Clarity point 1. As a result, I have updated my score.
> > >
> > > However, despite the detailed responses and my review of manuscript, I still find some aspects of the paper's clarity and methodology to be unclear. I acknowledge your approach in pointing at various segments of the paper in response to my questions, nonetheless, this does not alleviate my concerns and had hoped for an explanation.
> > >
> > > I do believe that your work addresses an intriguing problem: how do we ensure fairness in the actual outcome (treatment) rather than solely focusing on algorithmic recommendations through tailored suggestions to individuals. Despite this, I encounter difficulties in envisioning the paper, in its current state and with your responses, as being fully prepared for publication. I would be curious to hear the other reviewer's perspectives.

---

> > > > ### Author Response · Authors · 2023-08-17
> > > >
> > > > Thank you very much for your update!
> > > >
> > > > We apologize for the brevity: we'd prepared a longer response with more explanation, but were cut off by the 6000 character limit and wanted to get to all your points.
> > > >
> > > > Please, if there are further specific points we may clarify further, or if further explanations would be useful, would you mind listing them (perhaps referencing your original concerns or new ones?)?
> > > >
> > > > We are more than happy to explain further, but want to be concise and respect your time.
> > > >
> > > > Also, some explanations appeared in other responses: for example, in the global rebuttal we discussed extra exposition on assumptions. For example, about the Neyman Rubin potential outcomes framework:
> > > >
> > > > > In the standard Neyman-Rubin potential outcomes framework, individuals have a random vector of potential outcomes $Y(t)$ (indexed by treatment levels), but in observational data, under Assumption 1 (consistency and stable unit treatment values assumption) we only observe outcomes masked by the actual treatment assignment, aka $Y_i=Y(T_i)$. Under Assumption 3 (unconfoundedness) treatment is as-if randomized conditional on covariates. Assumption 3 is satisfied by design in randomized trials and otherwise is an assumption about the data-generating process, i.e. human decision-makers weren’t basing treatment decisions on unmeasured confounders unavailable in our dataset.
> > > >
> > > > In our rebuttal we included succinct versions of these additional explanations that could plausibly fit in our paper.
> > > >
> > > > Please note that many methods papers in causal inference/off-policy evaluation in Neurips typically include this kind of succinct explanation, rather than full textbook treatment of the framework and estimation (for that we refer to Imbens and Rubin, Hernan and Robins). We are happy to add explanatory sentences throughout but are running up against space constraints. We can also add some introductory exposition in plain-language:
> > > >
> > > > > In real data, we only observe causal outcomes for the treatments that were historically administered, although we want to assess different treatments and therefore estimate causal responses under different treatments. This is the so-called "fundamental problem of causal inference" and is modeled with the Neyman-Rubin potential outcomes framework. In this setting, it corresponds to the challenge that if, for example, we sent a coupon to a customer, in the database we only record how much they purchased after we sent that coupon, although we might be interested in estimating how much they purchase if we hadn't sent the coupon. Finally, our historical database reflects the selection bias of who we sent coupons to, which introduces confounding: selection bias introduces covariate shift of the population we sent coupons to previously vs. the population we are thinking about sending coupons to. A natural approach is to regress in the treated and control populations, and estimate population expectations under different treatment assignment rules. But, if our historical database reflected A/B tests (i.e. as-if-randomized data), we can further use information about the selection bias to design estimators that model the selection bias and use importance sampling to estimate counterfactuals. Finally, a long literature on improving estimators in causal inference can blend these different estimation approaches, which we describe later on in our special setting, to reduce variance.

---

### Official Review · Reviewer_QSPP · 2023-07-18

**Soundness:** 3 good
**Presentation:** 2 fair
**Contribution:** 3 good
**Rating:** 6
**Confidence:** 3

**Summary:**

This paper focuses on fair optimal decision rules, enhancing statistical estimators, and robustness checks for algorithmic recommendations with randomized decisions. It introduces a two-stage procedure with a complexity bound for optimizing within a constrained policy class, ensuring less conservative out-of-sample fairness constraint satisfaction.


**Strengths:**

1. The paper addresses an interesting topic, considering the randomness caused by humans in the loop and providing fairness guarantees. The technical aspects of the paper are robust and well-founded.

2. The paper explores two settings: one where R is randomized and satisfies overlap, and the other where R is deterministic but does not satisfy overlap. Comprehensive results are presented for both settings.

**Weaknesses:**

1. The fairness constraint considers the expectation of treatment but not recommendations which are the outcome of the algorithm. The reason for this is not clearly explained.

2. The assumptions made in the paper are not adequately cited or explained. While the author states that most assumptions are standard, some of them appear to be quite strong, such as assumption 5, which requires a strong decomposable linear function, and it's unclear if it can be satisfied in general.

3. There are some minor points that need clarification:
a. In Assumption 6, the symbol '\leq' may need to be changed to '\geq.'
b. Line 110 defines a cost function, but it seems like it should be referred to as a utility function, as indicated in line 157.
c. The conditions of each proposition are not clearly stated within the propositions, which causes some confusion, considering the numerous assumptions and settings.



**Questions:**

In proposition 2, L looks complicated. Is it explainable?

**Limitations:**

I may raise my score if the questions/weaknesses are properly explained.

---

> ### Author Rebuttal · Authors · 2023-08-08
>
> Thanks for your review! Below we make some clarifications, which we hope will help clear up that these are not weaknesses of the paper. We will add these explanations to clarify.
>
> Weaknesses
>
> - 1: See lines 40-46 which discuss scenarios where the fairness constraint should be on the expectation of treatment but not recommendation. Line 288 also quotes a report citing this motivation in the case study.
>
> Algorithmic auditing on the recommendation does not actually guarantee that there will not be disparities in realized treatment decisions, e.g. who actually finally redeems an offer or signs up for health insurance or receives preventive/punitive services. In a perfect world where treatment decisions by human decisions were optimal, there would be no difference in auditing recommendations vs. treatment decisions. But they are not, and this can be a central source of disparities; see e.g. discussion of administrative burden/cognitive burden affecting marginalized populations the most, making it more difficult to sign up for beneficial services, or due to discriminatory behavior of human decision makers. Imposing fairness constraints on treatment is a stronger guarantee of actual reductions in disparities.
>
> - 2. We will add an extra citation to causal inference textbooks here [21] (Hernan and Robins’ Causal Inference) and Imbens and Rubin [1] for the standard assumptions 1,3,6.
>
> [1] Imbens, Guido W., and Donald B. Rubin. Causal inference in statistics, social, and biomedical sciences. Cambridge University Press, 2015.
>
> Overall we propose to add the following sentences to the main text to further explain standard causal inference assumptions (we also included this in the rebuttal to all authors):
> > In the standard Neyman-Rubin potential outcomes framework, individuals have a random vector of potential outcomes $Y(t)$ (indexed by treatment levels), but in observational data, under Assumption 1 (consistency and stable unit treatment values assumption) we only observe outcomes masked by the actual treatment assignment, aka $Y_i=Y(T_i)$, and under Assumption 3 (unconfoundedness) treatment is as-if randomized conditional on covariates. Assumption 3 is satisfied by design in randomized trials and otherwise is an assumption about the data-generating process, i.e. human decision-makers weren’t basing treatment decisions on unmeasured confounders unavailable in our dataset.
>
> Also please note that in lines 128-142 we spend a lot of time explaining the non-standard assumptions specific to our paper.
>
> Finally, Re: Assumption 5: note that this doesn’t impose any functional form restrictions on the conditional cost function given covariates, i.e. $\mathbb{E}[c(Y)\mid T=1,X]$ is unrestricted. This just means that we can’t directly handle counterfactual cost functions, e.g. we rule out cases like $c(r,t,y) = T * Y(1-T)$. This is because of the fundamental problem of causal inference that the joint distribution of $(Y(1),Y(0))$ is unidentified and we don’t know what the counterfactual distribution of $P(Y(1) | T=0,X)$ is.
>
> For example, standard settings where firms get utility on the causal outcomes (i.e. revenue gained from a customer redeeming a 10 dollar off coupon and spending it) and some cost on the use of treatment (i.e. paying the 10 dollar discount) satisfy Assumption 5.
>
> So, Assumption 5 is relatively mild; it says that we have separate cost functions for treatment outcomes and causal $Y$ outcomes. Ultimately A5 is not very restrictive; we can handle nonstandard classification-type constraints/disparities by first applying the identification argument as in prop. 7 of the appendix and treating this as a covariate-conditional treatment cost.
>
> Thanks for your question on this; we’ll add this to the paper.
>
> 3. Thanks; that’s a minor typo, will be updated to $\nu_r, \nu_t \geq 0$. We use cost/utility interchangeably (difference is in sign). We list the assumptions together but will add which assumptions each prop. depends on. Prop 1,2,3 depend on A1-5. Prop 4 depends on A1-6. Prop 5,6 depend on A1-5 (no overlap). We will also clarify that all the propositions depend on A1-5 and what changes in different regimes is whether we assume overlap (A6) or not.
>
> Questions:
>
> - "L looks complicated... is it explainable?"
>
> Yes, it's explainable. Regression adjustment identification for causal inference (in Prop. 1) says that the optimal policy is to treat if $(\mu_1(X)-\mu_0(X)) (p_{1 \mid 1}(X)-p_{1 \mid 0}(X)) < 0$ (when outcomes are costs) if we didn't have any constraints. Think of this as a compliance-weighted CATE (conditional average-treatment effect), i.e. we weight $\tau$ (CATE =$\mu_1(X)-\mu_0(X)$) by the compliance effect of recommendations $ (p_{1 \mid 1}(X)-p_{1 \mid 0}(X))$. This is the unconstrained case. If we do have constraints, then Lagrange duality says that the optimal solution is given by the optimal unconstrained solution with an additional $\lambda$-multiplier on the constraint violations. The term $\left(p_{1 \mid 1}(X, A)-p_{1 \mid 0}(X, A)\right)\frac{\lambda}{p(A)}(\mathbb{I}[A=a]-\mathbb{I}[A=b])$ is just the integrand of the fairness constraint $\mathbb{E}[T(\pi) \mid A=a]-\mathbb{E}[T(\pi) \mid A=b]$ (by iterated expectations). So the $\lambda$ penalizes constraint violations. The term multiplying it is the contribution of a datapoint to estimating the constraint violation. This characterization of optimization over $\lambda$ is via Lagrange duality.
>
> We'll add this to the paper, thanks!

---

> > ### Author Response · Authors · 2023-08-16
> >
> > Thank you again for your questions. We want to follow up on the rebuttal. We understand that you may have a busy schedule, but we would greatly appreciate it if you could take a moment to review our response and provide any additional feedback.
> >
> > If you find our response useful, please consider updating the score to reflect the improvements made to the manuscript?
> >
> > Lastly, if there are any specific areas of concern that we can address or provide additional clarification on, please do not hesitate to let us know.

---

> > > ### Comment · Reviewer_QSPP · 2023-08-21
> > >
> > > Thank you very much for the clarifications and comments. I will take them into consideration during the discussion phase with AC.

---

### Official Review · Reviewer_2wiP · 2023-07-24

**Soundness:** 3 good
**Presentation:** 3 good
**Contribution:** 3 good
**Rating:** 6
**Confidence:** 1

**Summary:**

The authors characterize optimal and resource fairness-constrained optimal decision rules, and develop a doubly-robust estimator for the optimal decision rules.

**Strengths:**

I'm not very familiar with the areas of doubly-robust policy learning and am unable to assess the paper adequately.

**Weaknesses:**

The authors have not discussed the limitations of their work.

**Questions:**

See **Weaknesses** section.

**Limitations:**

See **Weaknesses** section.

---

> ### Author Rebuttal · Authors · 2023-08-09
>
> Thanks for the review! See global rebuttal where we propose to reiterate the limitations we discuss throughout the paper (e.g. our discussion right after assumptions) in a new concluding paragraph at the end.
>
>
> > “In summary, we provide theoretical characterization of fair encouragement designs with a human-in-the-loop, algorithms, and empirical demonstration. On the sociotechnical side, the limitations of our work include that interpreting any additional constraints implemented in our framework as improving fairness will depend on the context. On the technical side, our methodology is especially tailored to Assumption 6, about extrapolating responsivity to algorithms from the training data to the deployment environment. Although we can develop robustness checks to the violation of Assumption 6, this means we are really operating in one regime of “human-AI” collaboration and this method is not necessarily appropriate in all settings. Empirical work in different applications is required to verify the appropriateness of this assumption. Interesting directions for future work include algorithms that can handle intermediate settings, or use limited online learning to assess validity of assumptions.”

---

### Author Rebuttal · Authors · 2023-08-08

Thanks to all the reviewers for feedback and suggestions. We are encouraged that the reviewers find that we provide comprehensive theoretical/empirical results in tackling an important problem!

We respond point-by-point below and remark on some common points here. We believe these very minor  few sentences will improve clarity along these points. Thanks to the reviewers for identifying opportunities to clarify further.

- 2wiP and Cr9c note we don’t have a separate limitations paragraph. We propose to add the following concluding paragraph (which reiterates main limitations we discussed in the text, but explicitly labels them as such):
 > “*Conclusion and Limitations*: In summary, we provide theoretical characterization of fair encouragement designs with a human-in-the-loop, algorithms, and empirical demonstration. On the sociotechnical side, the limitations of our work include that interpreting any additional constraints implemented in our framework as improving fairness will depend on the context. On the technical side, our methodology is especially tailored to Assumption 6, about extrapolating responsivity to algorithms from the training data to the deployment environment. Although we can develop robustness checks to the violation of Assumption 6, this means we are really operating in one regime of “human-AI” collaboration and this method is not necessarily appropriate in all settings. Empirical work in different applications is required to verify the appropriateness of this assumption. Interesting directions for future work include algorithms that can handle intermediate settings, or use limited online learning to assess validity of assumptions.”

- Reviewers QSPP and Cr9c note that although we take care to carefully discuss non-standard assumptions specific to our paper, in the main text we don’t explain standard causal inference assumptions (consistency/SUTVA/unconfoundedness). This is a good point: assumptions are absolutely central to causal inference and they should be explained. As mentioned in specific reviewer response, we will add the following explanation of assumptions to the main text:

> "In the standard Neyman-Rubin potential outcomes framework, individuals have a random vector of potential outcomes $Y(t)$ (indexed by treatment levels), but in observational data. Under Assumption 1 (consistency and stable unit treatment values assumption) we only observe outcomes masked by the actual treatment assignment, aka $Y_i=Y(T_i)$, and under Assumption 3 (unconfoundedness) treatment is as-if randomized conditional on covariates. Assumption 3 is satisfied by design in randomized trials and otherwise is an assumption about the data-generating process, i.e. human decision-makers weren’t basing treatment decisions on unmeasured confounders unavailable in our dataset."

This is somewhat boilerplate for causal inference in general, which is why it wasn’t in the main text before; so adding this extra explanation doesn’t change anything about the paper.

Finally, Cr9c notes that there are opportunities to further include background material for readers unfamiliar with causal inference. Upon reflection, we recognize the writing is dense to fit our comprehensive results and we omit some background for readers unfamiliar with causal inference. But also in our response to Cr9c we note many areas where we have in fact included the information in question. We can certainly add additional background material on standard settings in the appendix, because we want to be as clear as possible to all audiences. Thanks to Cr9c for noting opportunities to improve clarity for audiences new to causal inference. On the other hand, multiple reviewers (h3Xk and xiA7) explicitly acknowledge the paper is well-written, and xiA7 even acknowledges the writing was already useful for someone new to causality. While we think adding additional emphases can help, we don’t at all think that the line edits that we included in our comprehensive response to Cr9c point to major technical flaws, and/or poor evaluation, limited impact, poor reproducibility and mostly unaddressed ethical considerations”.

We thank reviewers for the questions and noting opportunities for minor clarifications to further improve clarity of the camera-ready paper and we are absolutely confident that the minor adjustments we have laid out will do so in a camera-ready.

---

### Decision · Program_Chairs · 2023-09-21

**Decision:**

Accept (poster)

**Comment:**

The paper presents a fairness problem in algorithmic decision making, where different social groups may respond differently to the same (positive) decision: e.g., when given the same treatment recommendation, some groups are more likely to adhere to it than others, thereby creating different treatment outcomes, even though the decision made by the system/algorithm was the same. This leads to disparity in the actual benefits received by these groups.  The paper suggests a method to learn a decision policy that assigns recommendations to individuals so as to fulfill certain fairness criteria and reduce the disparity in the treatment realized.

The proposed problem was viewed by all reviewers as an interesting problem setting worth studying.  Although there is some disagreement on whether the paper's clarity and quality of writing is ready for publication, on balance the reviews are more positive than negative.